# Fast Intent Classification for LLM Routing via Statistical Analysis of Representations

## Abstract

Intent classification in Large Language Models (LLMs) involves categorizing user prompts into predefined classes. For instance, given a user prompt, the system must determine whether it primarily concerns mathematics, coding, or general text processing. Such classification enables routing prompts to specialized models optimized for specific domains, improving both accuracy and computational efficiency. In this work, we conduct a systematic study comparing training-free vs training-based approaches for intent classification. For this purpose, we introduce two lightweight, training-free methods based on statistical analysis of internal model representations and compare them against MLP classifiers and linear probes. The training-free methods use key statistical metrics from hidden features, enabling intent inference during the initial forward pass with minimal overhead. Our comprehensive empirical evaluation reveals that 1) both training-free and training-based methods saturate easy benchmarks (mathematics vs. coding vs. natural language), 2) Training-based classifiers have an advantage on harder classification tasks (e.g. Java vs Python), and 3) Training-free methods are more robust to out-of-distribution and ambiguous prompts.

## 1 Introduction

Large Language Models (LLMs) have demonstrated remarkable capabilities across diverse tasks including mathematical reasoning (Wei et al., 2022; Yao et al., 2023; Gao et al., 2023) and code generation (Li et al., 2022; Guo et al., 2024; Zhu et al., 2024). While many LLMs are trained for general purpose tasks (Radford et al., 2019; Raffel et al., 2020), current state-of-the-art is moving towards a routing approach where an intent classifier is used to detect user intent and then send the prompt to a specific model (OpenAI, 2025; Bocklisch et al., 2017; Bunk et al., 2020; Arora et al., 2024). This has the benefit of improving inference efficiency of production-scale LLM systems. Approaches to intent classification either rely on LLM calls, which is prone to hallucination (Bang et al., 2023; Banerjee et al., 2025), or on dedicated classification models, which requires extensive training data and computational resources Larson et al. (2019); Chen et al. (2019). In this work, we conduct an empirical study to evaluate advantages and disadvantages of training-free methods, as compared to training-based ones. For this purpose, we introduce two training-free, methods for intent classification, VecStat and NormStat, that operate entirely in the prefill phase with negligible extra cost. The motivation is the observation that different prompt types (mathematics, coding, general text, etc.) induce distinct activation distributions. Specifically, VecStat and NormStat represent two levels of statistical compression: VecStat preserves directional information but induces higher storage and calibration cost, while NormStat aggregates radial evidence and enjoys minimal memory consumption. Theoretical analysis further clarifies this trade-off: VecStat is preferable when class differences are primarily directional, whereas NormStat suffices—and is more memory-efficient—for isotropic-scale separation (coarse-grained classification tasks) thanks to the dimension-free calibration complexity.

Beyond computational efficiency relative to direct LLM calls, a key benefit of statistical methods is uncertainty quantification. With a one-line softmax normalization, statistical methods yield well-calibrated class probabilities, including on mixed-intent prompts; In contrast, we show that a training-based method using MLP head (inspired from sentence-classification pipelines (Casanueva et al., 2020; Jiang et al., 2024)) typically requires post-hoc calibration (e.g., temperature scaling) to avoid overconfidence (Guo et al., 2017). When classes change, our methods also support rapid,

| Method | FLOPs Overhead | Memory Overhead | Extendability of New Classes | Uncertainty Quantification |
|--------|----------------|-----------------|------------------------------|----------------------------|
| **NormStat** | $O(Td)$ | $O(m)$ | Compute new baselines | Works well |
| **VecStat** | $O(Td)$ | $O(md)$ | Compute new baselines | Works well |
| **MLP** | $O(hd)$ | $O(hd)$ | Retrain a new MLP head | Possible but needs extra calibration |
| **LLM Call** | $\Omega(Td^2)$ | $\sim$ | Extend via prompt engineering | $\times$ |

Table 1: Comparison of classifiers. $T$: prompt length; $d$: hidden width; $m$: # classes; $h$: MLP hidden size.

incremental updates by simply appending new class statistics, avoiding any retraining. Table 1 compares these methods by compute, memory, extendability, and uncertainty quantification.

Hereafter, we use the wording training-free vs training-based methods to refer to whether gradient updates are needed. We conduct extensive empirical analysis to compare the performance of VecStat and NormStat against the training-based MLP classifier applied at the LLM's final projection layer. The MLP head is trained on labeled data with Adam, whereas our statistical methods require no gradient updates. We apply these methods to LLMs ranging from 1B to 32B parameters and evaluate intent classification at both coarse-grained and fine-grained levels across seven benchmark datasets. The empirical results reveal that *there is no one-fits-all model for intent classification*. On the one hand, NormStat, VecStat, and MLP classifiers saturate easy benchmarks. On the other hand, MLP classifiers typically achieve higher accuracy on hard tasks, yet they suffer from overconfidence in predictions, limiting their ability to provide reliable predictions on ambiguous prompts, a scenario where NormStat seems to outperform other methods. Our contributions are:

- We introduce NormStat and VecStat, two training-free statistical methods that perform intent classification directly within the LLM prefill phase, requiring $O(Td)$ additional computation compared to $\Theta(Td^2)$ forward pass cost, enabling deployment with negligible latency overhead.

- We provide theoretical analysis showing when each method excels: VecStat performs best when prompt types differ in feature directions, while NormStat is optimal when they differ in overall magnitude, with NormStat requiring fewer calibration samples to achieve comparable accuracy.

- We conduct a comprehensive empirical study across seven LLMs (1B-32B parameters) on both coarse-grained and fine-grained intent classification tasks, comparing training-free methods vs training-based ones. Our results reveal that both training-free and training-based methods excel at easy tasks, saturating the benchmarks. Training-free methods provide superior uncertainty quantification for mixed-intent prompts compared to training-based approaches, while training-based methods are better at more fine-grained tasks.

## 1.1 RELATED WORK

**Task Classification** Intent classification maps a user prompt to a predefined label. Classical approaches either (i) train supervised classifiers over tokenized utterances (e.g., CNNs) to produce a distribution over intents (Hashemi et al., 2016; Goo et al., 2018; He et al., 2019), or (ii) fine-tune contextual encoders, particularly BERT-based models, where hidden states feed specialized intent classification heads, often jointly trained with slot filling tasks (Chen et al., 2019; Bocklisch et al., 2017; Bunk et al., 2020). In modern LLM-based systems, intent classification serves as a critical routing mechanism that allows the selection of appropriate downstream tools and models, enforces guardrails and fallback policies, and optimizes inference cost and latency (Souha et al., 2023; Arora et al., 2024). The predominant approach involves direct LLM inference through several key techniques (Liu et al., 2023; Rodriguez et al., 2024; Wang et al., 2023b; Arora et al., 2024; Hong et al., 2024; Wei et al., 2022). However, the computational expense of LLM inference at scale has motivated hybrid architectures that combine fast, lightweight classifiers (including PEFT-tuned encoders) with LLMs through uncertainty-aware routing mechanisms. These systems employ confidence thresholding, entropy-based measures, or learned routing policies to reserve expensive LLM calls for ambiguous cases where simpler models exhibit high uncertainty (Liu et al., 2022; 2024).

**LLMs as text encoder** Recent advances in LLMs have prompted researchers to explore their use as text encoders. An interesting approach is embedding extraction where existing methods typically operate on the last layer outputs through three strategies: using the last token embedding (Ma et al., 2024; Neelakantan et al., 2022; Wang et al., 2024; Meng et al., 2024; Jiang et al., 2024), averaging across all token embeddings (Muennighoff, 2022; Muennighoff et al., 2024; BehnamGhader et al.,

2024), or employing trainable modules (Lee et al., 2024; Tang & Yang, 2024). Interested readers can refer to (Tao et al., 2024; Nie et al., 2024) for a more detailed review on this topic. In contrast to these approaches, this work addresses user-intent classification for routing where both accuracy and computational efficiency are primary considerations. Our method utilizes prefill-time outputs from general-purpose LLMs without modification or additional training. By leveraging computational intermediates already produced during LLM prefill phase, this approach avoids the storage overhead of maintaining a dedicated billion-parameter model for intent classification.

**Neural Feature Analysis** Our approach extracts representations $Wz$, where $W$ is a pretrained weight matrix and $z$ is model's hidden state. This design is motivated by two lines of research. First, linear probes effectively extract semantic information from transformer representations (Alain & Bengio, 2016; Hewitt & Manning, 2019), with sparse autoencoder studies suggesting that many concepts are captured by a small number of sparse features in the activation space (Cunningham et al., 2024; Gao et al., 2024). Superposition theory provides theoretical grounding, explaining how features remain recoverable through linear projections (Elhage et al., 2022). Second, activation steering research demonstrates that intent-related behaviors can be manipulated through linear interventions in the representation space (Turner et al., 2023; Panickssery et al., 2023). Finally, activations $Wz$ were successfully used in Hayou et al. (2025) to determine target module for LoRA finetuning, showing that activation capture data signal.

## 2 Methodology

In Large Language Models, prefill refers to the first forward pass of the user prompt. During this time, KV-cache is filled for autoregressive decoding (Shazeer, 2019; Ainslie et al., 2023; Chang et al., 2024; Aguirre et al., 2025; Jie et al., 2025) and gets updated for each token generation. In the prefill, we already compute the prompt's first forward pass to build keys/values; adding a light classifier there adds low cost but could provide a strong signal to route the prompt if needed: simple prompts remains on a small, cheap model; math/code/reasoning prompts routed to a larger or specialized model. Making this decision before the first generated token avoids wasting computation on the mismatched model. This is even more important if routing is customized for each user.

Modern LLM serving imposes a *lightweight* constraint on any prefill-time intent classifier used for routing: (i) the classifier's extra computation must be negligible compared to a single forward pass, and (ii) per-prompt memory and persistent storage must be negligible. Concretely, for a prompt of length $T$, a forward pass cost $\Theta(Td^2)$[1] computation in any given layer with hidden dimension $d$; a *lightweight* classifier should at most add $o(Td^2)$ cost, ideally $O(Td)$. Likewise, per-request state must be $O(1)$–$O(d)$ floats (not $O(Td)$), and per-class baselines must be $O(1)$–$O(d)$ numbers. More details are provided later in the paper.

In the following, we introduce a training-free approach to intent classification, based on a statistical analysis of hidden features in LLMs, and satisfies the computational constraints above. Figure 1 summarizes the pipeline and contrasts between the two proposed methods VecStat with NormStat. In general, our method (i) collects module features; (ii) computes summary statistics; (iii) compares these statistics to per-class baselines via Gaussian KL (or cosine distance for VecStat); (iv) averages distances over layers and applying a softmax over the negative distances to obtain class probabilities and a routing decision.

### 2.1 A Statistical Approach to Intent Classification

Consider an LLM with weight modules $\mathcal{M} = \{W_1, W_2, \ldots, W_p\}$, for some $p \geq 1$. The weights modules $\mathcal{M}$ represent all available weight matrices in the model, across layer index and module type. We will abuse the notation and use $W_\ell$ to refer to both the module and its weight matrix.

Let $\mathbf{x} = (x_t)_{1 \leq t \leq T}$ be a prompt of $T$ tokens. For each weight module $W_k$, let $(y_{\ell,t})_{1 \leq t \leq T}$ denote the output features in module $W_\ell$. For instance, $(y_{\ell,t})_{1 \leq t \leq T}$ could be the output of a Query head, or

---

[1]The forward pass cost is $\Theta(Td^2 + T^2d)$. In the regime $d > T$, the $Td^2$ term dominates, so we drop the $T^2d$ term and write the cost as $\Theta(Td^2)$; retaining $T^2d$ does not affect our conclusions.

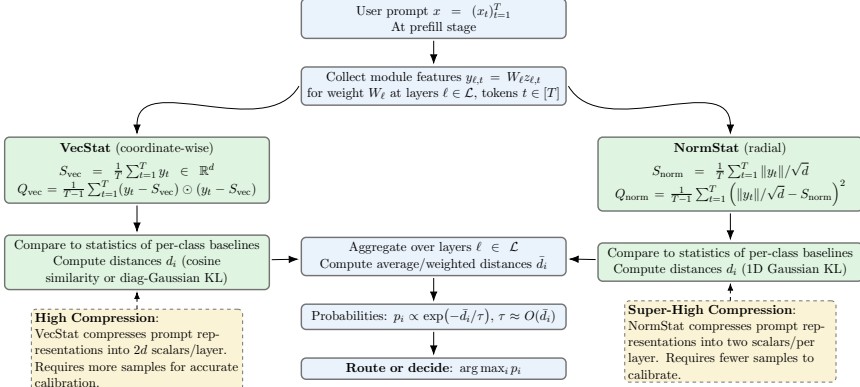

Figure 1: Flowchart of VecStat vs NormStat for prefill-time intent classification and routing.

the projection layer in an MLP block. Each $y_{\ell,t}$ is a $d$-dimensional vector given by $y_{\ell,t} = W_\ell z_{\ell,t}$, where $d$ is the output dimension in module $W_\ell$, and $z_{\ell,t}$ is the input to that module for token $t$.

**Intent classification.** We aim to classify the prompt $\mathbf{x}$ into one of the classes $C_1, C_2, \ldots, C_m$, where $m \geq 2$. For instance, a binary classification where $C_1$ is mathematics and $C_2$ is coding. For each module $W_\ell$, we compute lightweight summary statistics from the features $\{y_{\ell,t}\}_{t=1}^T$ and compare them to per-class baselines: (i) for each class $C_i$, precompute the same statistics on *calibration data* at the same module $W_\ell$; (ii) for the incoming prompt, compute the statistics at $W_\ell$ and measure similarity to each baseline; (iii) predict the class $C_i$ with the highest similarity.

Several statistics are considered, such as the mean and covariance of $\{y_{\ell,t}\}_{t=1}^T$, to capture geometric information about hidden features. Estimating the mean requires $O(Td)$ calculations while the covariance requires $O(Td^2)$. Thus, using covariance is not computationally efficient, since it violates the $O(Td)$ condition above. However, we consider a weaker variant where we only estimate coordinate-wise variance (diagonal of the covariance matrix). We call this method VecStat, which relies on coordinate-wise mean and variance for classification. We also introduce a lighter weight method called NormStat, which relies solely on the norm statistic across all tokens and coordinates.

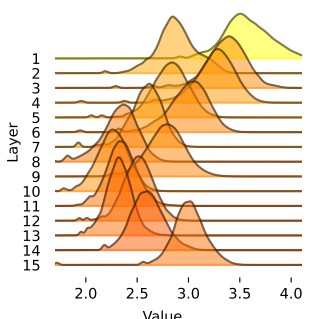

Figure 2: **Per-token query-norm distributions across layers.** Histograms of $\{\|y_{\ell,t}\|/\sqrt{d}\}_{t=1}^T$ for representative layers $\ell$. Shapes are close to Gaussian with layer-dependent mean/variance, which supports a 1D Gaussian proxy for NormStat.

In the following, we present the two methods in the single-layer case. When multiple layers are used, we aggregate similarity scores across $\ell$ by averaging.

**Vector Statistic** (VecStat): calculate *coordinate-wise* token means and second moments tokens

$$S_{\text{vec}} = \frac{1}{T} \sum_{t=1}^T y_t \in \mathbb{R}^d, \qquad Q_{\text{vec}} = \frac{1}{T-1} \sum_{t=1}^T (y_t - S_{\text{vec}}) \odot (y_t - S_{\text{vec}}) \in \mathbb{R}^d. \qquad (1)$$

**Norm Statistic** (NormStat): summarize each $y_t$ through a the norm $\|y_t\|$ and aggregate across tokens to obtain the statistics

$$S_{\text{norm}} = \frac{1}{T} \sum_{t=1}^T \frac{\|y_t\|}{\sqrt{d}} \in \mathbb{R}, \quad Q_{\text{norm}} = \frac{1}{T-1} \sum_{t=1}^T \left( \frac{\|y_t\|}{\sqrt{d}} - S_{\text{norm}} \right)^2 \in \mathbb{R}. \qquad (2)$$

With both methods, we use closed-form Gaussian KL divergence (5) and (6) to measure the similarity between prompt and classes' statistics. Intuitively, this acts as a proxy for the true KL-divergence between distributions which is prohibitively expensive to compute. Across models and datasets, we observed that the radial token features $\|y_{\ell,t}\|/\sqrt{d}$ are Gaussian-like for fixed $\ell$ (Fig. 2), and similar observations hold for the coordinates of $y_t$. [2]

---

[2]One could estimate an empirical KL without summaries, but doing so robustly at inference time is prohibitively expensive in both compute and memory.

These two choices form a *statistical compression ladder*: VecStat keeps per-coordinate first and second moments, while NormStat compresses all coordinates to a single *radial* information per token and then to its mean/variance across tokens.

The rest of this section develops this story rigorously: (i) we prove when each is statistically preferable using a simplified Gaussian setting, and (ii) we connect those guarantees to *compute*, *memory*, and *calibration* costs. All the proofs are deferred to Appendix B.

## 2.2 INTUITIVE ANALYSIS IN A GAUSSIAN SETTING

Intuitively, VecStat retains more information about feature distribution than NormStat because it tracks coordinate-wise statistics instead of a single statistic for each module. To understand the difference between these two methods, we consider a Gaussian setting with diagonal covariance and study regimes where NormStat is competitive with VecStat, in which case NormStat is preferred for computational efficiency. Additional analysis is in Appendix A.

**Setting and notation.** Here we study general features $(y_t)_{1 \le t \le T}$ (not necessarily representations in an LLM). For each baseline class $k \in \{1, \ldots, m\}$ and tokens $t = 1, \ldots, T$, assume that

$$y_t \mid k \sim \mathcal{N}(\mu_k, \Sigma_k) \subset \mathbb{R}^d, \qquad \text{independently across } t, \tag{3}$$

where, $\Sigma_k = \mathrm{Diag}(\sigma_{k,1}^2, \ldots, \sigma_{k,d}^2)$ is a diagonal covariance matrix.

NormStat observes only the norm $\{\|y_t\|\}$ and is invariant to rotations of the features; it cannot detect separation that lives in direction. In contrast, VecStat retains per-coordinate first/second moments and is therefore rotation-sensitive. In an *isotropic-scale* setting (equal means, spherical covariances with different variance), the optimal likelihood ratio reduces to a monotone function of the total radial sum $\sum_t \|y_t\|^2$, so NormStat is already Bayes-optimal. But in a *directional* setting (equal covariance, equal mean norms, different mean directions), every radius-only rule is blind, whereas a coordinate-aware test achieves exponentially small error in $T$. The next theorem formalizes this.

**Theorem 1.** *[NormStat vs VecStat] Assume a binary classification $k \in \{1, 2\}$ with uniform prior.*

1. ***Directional regime (NormStat×, VecStat✓).*** *Assume $\Sigma_1 = \Sigma_2 = \sigma^2 I_d$, $\|\mu_1\| = \|\mu_2\|$, and $\mu_1 \neq \mu_2$. Then any classifier whose decision depends only on the norms $\{\|y_t\|\}_{t=1}^T$ has Bayes error $1/2$. Moreover, the likelihood-ratio test achieves error probability*

$$\Pr(\hat{k} \neq k) \le \exp\left(-\frac{T}{8\sigma^2}\|\mu_1 - \mu_2\|^2\right), \quad \text{where } \hat{k}(y_{1:T}) := \begin{cases} 1, & \langle S_{\mathrm{vec}}, \mu_1 - \mu_2 \rangle \ge 0, \\ 2, & \text{otherwise.} \end{cases}$$

2. ***Isotropic-scale regime (NormStat✓, VecStat ties).*** *Assume $\mu_1 = \mu_2 = 0$ and $\Sigma_k = \sigma_k^2 I_d$ with $\sigma_1 \neq \sigma_2$. Then the Log-Likelihood Ratio is a strictly monotone function of the radial statistic $R_T := \sum_{t=1}^T \|y_t\|^2$; hence every Bayes-optimal test depends only on $R_T$, and adding coordinate-wise information cannot improve its Bayes risk.*

Theorem 1 isolates two extremes, whereas real prompts generally result in a mix between these extremes. An example is the following: assume $y_t$ are i.i.d. from a sign–mixture with $s_t \in \{\pm 1\}$,

$$y_t \sim \pi \mathcal{N}(+\mu, \sigma^2 I_d) + (1 - \pi)\mathcal{N}(-\mu, \sigma^2 I_d), \quad \gamma := \mathbb{E}[s_t] = 2\pi - 1,$$

where $\mu \in \mathbb{R}^d$, $\pi \in (0, 1)$. Hence,

$$\mathbb{E}[S_{\mathrm{vec}}] = \gamma \mu, \qquad \mathbb{E}[Q_{\mathrm{vec}}] = \sigma^2 \mathbf{1}_d + (1 - \gamma^2)\,\mu \odot \mu, \qquad \mathbb{E}[S_{norm}] = \mathbb{E}\|y_t\| = F(\mu, \sigma^2),$$

where $F(\mu, \sigma^2)$ depends only on $(\mu, \sigma^2)$. For NormStat, since norms are even, the distribution of $\|y_t\|$ is invariant under the $\pm\mu$ mixture. Consequently, the distributions of $S_{\mathrm{norm}}$ and $Q_{\mathrm{norm}}$ do not depend on $\pi$. As $\pi \to \frac{1}{2}$, the advantage of VecStat over NormStat shrinks; at $\pi = \frac{1}{2}$, the mean component $S_{\mathrm{vec}}$ cancels, and VecStat effectively reduces to its second–moment part, aligning with the radial evidence summarized by NormStat. Away from $\frac{1}{2}$, $S_{\mathrm{vec}}$ provides a clear benefit.

**Calibration Cost.** An important aspect of statistical methods is sample complexity, or more specifically, the convergence rate in the number of samples. This provides an estimate of the total number of calibration samples needed to create the classes $k \in \{1, 2, \ldots, m\}$. The next theorem show the calibration advantage of NormStat over VecStat.

**Theorem 2** (Calibration cost). *Fix a class $k$. Let $y_1, \ldots, y_N \overset{i.i.d.}{\sim} \mathcal{N}(\mu_k, \Sigma_k)$ in $\mathbb{R}^d$, where $N$ is the number of calibration samples drawn for this class. Let $q = \mathbb{E}[\|y_1\|]$ and define $\hat{\mu}_k = N^{-1} \sum_{i=1}^{N} y_i$, and $\quad \hat{q} = N^{-1} \sum_{i=1}^{N} d^{-1/2} \|y_i\|$. Then, for any $\delta \in (0, 1)$, with probability at least $1 - \delta$, we have:*

1. ***NormStat (dimension-free):*** $|\hat{q} - q| \lesssim \sqrt{\frac{\log(1/\delta)}{N}}$.

2. ***VecStat (dimension-dependent):*** $\|\hat{\mu}_k - \mu_k\|_2 \lesssim \sqrt{\frac{d + \log(1/\delta)}{N}}$.

Considering just the statistics $\hat{\mu}$ and $\hat{q}$, to obtain an estimation error of order $\epsilon$, one needs $N = \Omega(\epsilon^{-2})$ for $\hat{q}$ and $N = \Omega(d\epsilon^{-2})$ for $\hat{\mu}$, showing the computational advantage of NormStat over VecStat. This is particularly important in data scarce regimes with few samples for each class. We discuss this in more details in the next section.

Theorem 1 and Theorem 2 compare our methods from two different angles: (i) expressivity, where VecStat has an edge if directional information is important, otherwise NormStat ties with VecStat, (ii) calibration cost, where NormStat has an edge with fewer calibration samples needed to reach a given error level. A third important angle is storage/memory cost: while both methods have similar classification cost ($O(Td)$ per module), NormStat uses $O(B)$ scalars and $O(B)$ scoring FLOPs, while VecStat uses $O(Bd)$ numbers and $O(Bd)$ scoring FLOPs—so for large $B$ or tight memory/latency budgets, NormStat has an advantage.

## 2.3 TRAINING-BASED INTENT CLASSIFICATION

For a comprehensive empirical study, we consider an intent classifier based on a trained head on top of frozen LLM features. We use the last Transformer block and write $y_{\ell^\star,t} \in \mathbb{R}^d$ for its token features ($t = 1, \ldots, T$). Inspired by prompt/sentence classification pipelines (e.g., (Ma et al., 2024; Wang et al., 2024; Meng et al., 2024)), we build a single prompt-level vector in two ways:

$$\textbf{Avg-MLP:} \quad z_{\text{avg}} := \frac{1}{T} \sum_{t=1}^{T} y_{\ell^\star,t} = S_{\text{vec}}^{(\ell^\star)} \in \mathbb{R}^d, \qquad \textbf{Tail-MLP:} \quad z_{\text{tail}} := y_{\ell^\star,T} \in \mathbb{R}^d.$$

Given $z \in \{z_{\text{avg}}, z_{\text{tail}}\}$, we train a two-layer MLP with hidden width $h$ with cross entropy loss. Since $z_{\text{avg}}$ or $z_{\text{tail}}$ is produced during prefill, the incremental latency is a single MLP forward pass. We also consider a simple linear probe variant that we call **Avg-Linear** where we use a simple projection instead of MLP.

## 3 EXPERIMENTS

In this section, we evaluate the effectiveness of NormStat, VecStat, Avg-MLP, Tail-MLP, and Avg-Linear across multiple LLMs and classification datasets.[3] Comprehensive experimental details can be found in Appendix C, and additional experimental results are presented in Appendix D. Our results provide a systematic comparison between training-free and training-based intent classification methods, showing the advantages and disadvantages of each approach. For completeness, we also compare with a method relying on direct LLM call for intent classification.

## 3.1 EXPERIMENTAL SETUP

**Classification granularities** We consider two levels of granularity. Level-1 addresses coarse-grained classification across three domains: general text, mathematics, and code. Level-2 tests fine-grained separation within each domain: identifying programming languages, mathematical subfields, natural languages. For Level-1 calibration, we use representative datasets: MMLU European

---

[3]Source code is provided in the supplemental materials

History (Hendrycks et al., 2021a;b) for general text, GSM8K (Cobbe et al., 2021) for mathematics, and Magicoder (Wei et al., 2023) for code. We test on MMLU US History for general text, GSM8K and MATH500 (Lightman et al., 2023) for mathematics (in-distribution and out-of-distribution respectively), and Magicoder and HumanEval (Chen et al., 2021) for code (in/out of distribution). Level-2 experiments utilize domain-specific subsets: Magicoder for programming language identification, Competition Math (Hendrycks et al., 2021c) for mathematical subfield classification, and the Aya dataset (Singh et al., 2024) for natural language identification, with each dataset split between calibration and test sets. For more details, please refer to Appendix C.1.

**Method and LLM selection** We compare five classification methods: NormStat, VecStat (with two variants: cosine similarity, VecStat:Cos, and KL divergence, VecStat:KL), and training-based baselines Avg-MLP, Tail-MLP, and Avg-Linear. Training-based methods use the same calibration data for training to ensure fair comparison. We also benchmark LLM calls, where model predicts intent directly from the prompt. We evaluate both zero-shot and 3-shot in-context learning (ICL) settings, with the 3-shot variant providing one example per class (see Table 7). Our testing reveals that providing a high-level overview of the intent classes in the prompt is necessary for achieving a reasonable performance.[4] All calibration prompts are truncated to 512 tokens, with

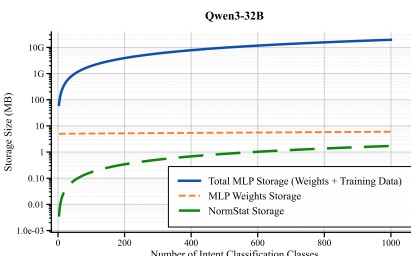

Figure 3: Storage simulation for intent classification on Qwen3-32B as the number of classes increases to 1,000.

training-free methods probing all linear modules. We evaluate on Qwen3 and Llama families, spanning 1B to 32B scales, and base and instruction-tuned variants (see Appendix C.3 for more details).

**Evaluation Metrics** We compute accuracy on each test dataset independently, where each dataset contains samples from a single ground-truth class. This approach ensures our evaluation is not biased by varying dataset sizes across classes. For Level-2 classification, we report mean accuracy across all classes within each task due to space constraints. This mean accuracy corresponds to the balanced accuracy metric, providing equal weight to each class regardless of test set size and effectively handling the natural class imbalance among test datasets. All experiments use three random seeds, and we report mean performance with standard deviation.

**Training/Calibration Cost Comparison.** Both training-based and training-free methods add negligible latency to intent classification and their computational overhead is minimal compared to the LLM prefill phase. The fundamental difference lies in how adaptable these methods are with evolving classification targets. When adding a new class, training-free methods maintain constant cost: *computing statistics for the new class data*. On the contrary, training-based methods face a trade-off: either (i) regenerate training embeddings for all classes (both existing and new) from scratch to enable retraining—a process whose cost grows linearly with the total number of classes, or (ii) cache all training embeddings to enable rapid retraining at the expense of storage. Fig. 3 quantifies the storage implications of option (ii) for Qwen3-32B: as the number of classes gradually increases to 1,000, the total storage requirement reaches approximately 20GB (primarily for cached embeddings), although the MLP weights alone require only 6MB. By comparison, NormStat requires only 1.7MB for the same 1,000 classes—a difference of over an order of magnitude. As a result, *training-free methods are particularly well-suited for dynamic intent classification* systems where new classes are frequently added or removed.

## 3.2 EMPIRICAL RESULTS

Table 2 reports classification results at both granularity levels for five methods on four representative LLMs, while Table 3 provides detailed per-class accuracy for level-2 programming language classification. Results for all seven LLMs are reported in Appendix D.1 (level-1) and D.2 (level-2).

---

[4]Due to the complexity of designing effective prompts for level-2 finer-grained distinctions, we evaluate direct LLM inference only on Level-1 classification tasks.

Table 2: Classification accuracy across eight methods on level-1 and level-2 classification granularities. Level-1 reports per-dataset accuracy for coarse-grained domain classification (general text, math, code). Level-2 reports average accuracy across classes within each domain: programming languages, mathematical subfields, and natural languages. LLM Call results omitted for Llama-3.2-1B models (failed to produce valid outputs).

| Model | Method | Level-1 | | | | | Level-2 | | |
|---|---|---|---|---|---|---|---|---|---|
| | | gsm8k | humaneval | magicoder | math500 | mmlu_history | code | math | natural language |
| Qwen3-1.7B | Avg-MLP | $99.97_{\pm0.04}$ | $99.80_{\pm0.35}$ | $99.99_{\pm0.02}$ | $78.33_{\pm1.67}$ | $100.00_{\pm0.00}$ | $99.96_{\pm0.01}$ | $76.77_{\pm0.27}$ | $99.93_{\pm0.02}$ |
| | Tail-MLP | $100.00_{\pm0.00}$ | $100.00_{\pm0.00}$ | $99.97_{\pm0.02}$ | $99.00_{\pm0.00}$ | $100.00_{\pm0.00}$ | $99.59_{\pm0.02}$ | $72.12_{\pm0.27}$ | $99.87_{\pm0.05}$ |
| | Avg-Linear | $100.00_{\pm0.00}$ | $99.80_{\pm0.35}$ | $100.00_{\pm0.00}$ | $69.20_{\pm9.18}$ | $100.00_{\pm0.00}$ | $99.96_{\pm0.01}$ | $74.62_{\pm0.24}$ | $99.94_{\pm0.02}$ |
| | NormStat:KL | $97.35_{\pm0.00}$ | $70.12_{\pm0.61}$ | $99.27_{\pm0.11}$ | $76.60_{\pm0.20}$ | $100.00_{\pm0.00}$ | $57.79_{\pm0.73}$ | $38.39_{\pm0.56}$ | $89.77_{\pm0.25}$ |
| | VecStat:KL | $100.00_{\pm0.00}$ | $99.39_{\pm0.00}$ | $99.97_{\pm0.03}$ | $88.33_{\pm0.30}$ | $100.00_{\pm0.00}$ | $99.24_{\pm0.12}$ | $48.36_{\pm0.64}$ | $99.20_{\pm0.08}$ |
| | VecStat:Cos | $100.00_{\pm0.00}$ | $98.78_{\pm0.06}$ | $99.97_{\pm0.03}$ | $92.26_{\pm0.11}$ | $100.00_{\pm0.00}$ | $99.08_{\pm0.16}$ | $52.74_{\pm0.67}$ | $99.57_{\pm0.02}$ |
| | LLM Call (0-shot) | $90.60_{\pm0.00}$ | $0.00_{\pm0.00}$ | $81.03_{\pm0.39}$ | $59.40_{\pm0.00}$ | $4.41_{\pm0.00}$ | - | | |
| | LLM Call (3-shot) | $64.54_{\pm0.88}$ | $33.54_{\pm3.23}$ | $85.71_{\pm0.26}$ | $80.87_{\pm1.10}$ | $58.17_{\pm3.93}$ | - | | |
| Llama-3.2-1B | Avg-MLP | $100.00_{\pm0.00}$ | $100.00_{\pm0.00}$ | $99.99_{\pm0.01}$ | $64.33_{\pm8.33}$ | $99.84_{\pm0.28}$ | $99.97_{\pm0.01}$ | $70.20_{\pm0.72}$ | $99.91_{\pm0.04}$ |
| | Tail-MLP | $99.95_{\pm0.04}$ | $100.00_{\pm0.00}$ | $99.97_{\pm0.00}$ | $98.93_{\pm0.42}$ | $99.35_{\pm1.13}$ | $99.47_{\pm0.07}$ | $69.01_{\pm1.11}$ | $99.83_{\pm0.07}$ |
| | Avg-Linear | $100.00_{\pm0.00}$ | $99.80_{\pm0.35}$ | $99.99_{\pm0.01}$ | $71.47_{\pm7.78}$ | $98.69_{\pm1.86}$ | $99.96_{\pm0.00}$ | $68.87_{\pm1.06}$ | $99.90_{\pm0.02}$ |
| | NormStat:KL | $99.49_{\pm0.09}$ | $90.85_{\pm0.00}$ | $96.39_{\pm0.20}$ | $83.40_{\pm0.00}$ | $92.48_{\pm0.57}$ | $49.22_{\pm0.73}$ | $31.27_{\pm0.54}$ | $86.53_{\pm1.33}$ |
| | VecStat:KL | $100.00_{\pm0.00}$ | $99.39_{\pm0.00}$ | $99.98_{\pm0.02}$ | $78.80_{\pm0.12}$ | $100.00_{\pm0.00}$ | $98.99_{\pm0.10}$ | $47.67_{\pm0.89}$ | $99.19_{\pm0.08}$ |
| | VecStat:Cos | $100.00_{\pm0.00}$ | $99.39_{\pm0.00}$ | $99.97_{\pm0.02}$ | $77.60_{\pm0.20}$ | $100.00_{\pm0.00}$ | $98.71_{\pm0.12}$ | $51.47_{\pm0.73}$ | $99.72_{\pm0.01}$ |
| Qwen3-8B | Avg-MLP | $99.42_{\pm0.74}$ | $99.59_{\pm0.35}$ | $99.99_{\pm0.02}$ | $77.27_{\pm11.02}$ | $100.00_{\pm0.00}$ | $99.97_{\pm0.01}$ | $79.10_{\pm0.48}$ | $99.92_{\pm0.01}$ |
| | Tail-MLP | $99.82_{\pm0.12}$ | $98.58_{\pm2.46}$ | $99.98_{\pm0.02}$ | $81.20_{\pm9.72}$ | $100.00_{\pm0.00}$ | $99.67_{\pm0.01}$ | $75.76_{\pm0.68}$ | $99.88_{\pm0.04}$ |
| | Avg-Linear | $99.57_{\pm0.24}$ | $100.00_{\pm0.00}$ | $100.00_{\pm0.00}$ | $83.00_{\pm2.03}$ | $100.00_{\pm0.00}$ | $99.97_{\pm0.01}$ | $77.70_{\pm1.11}$ | $99.94_{\pm0.01}$ |
| | NormStat:KL | $85.14_{\pm0.35}$ | $10.37_{\pm1.06}$ | $99.85_{\pm0.06}$ | $92.93_{\pm0.12}$ | $99.51_{\pm0.49}$ | $56.39_{\pm0.69}$ | $33.25_{\pm0.92}$ | $90.09_{\pm0.40}$ |
| | VecStat:KL | $99.95_{\pm0.04}$ | $99.59_{\pm0.35}$ | $99.99_{\pm0.02}$ | $92.20_{\pm0.00}$ | $100.00_{\pm0.00}$ | $99.23_{\pm0.10}$ | $49.99_{\pm1.12}$ | $99.20_{\pm0.08}$ |
| | VecStat:Cos | $100.00_{\pm0.00}$ | $95.73_{\pm0.61}$ | $99.98_{\pm0.03}$ | $94.20_{\pm0.00}$ | $100.00_{\pm0.00}$ | $99.34_{\pm0.09}$ | $54.75_{\pm0.63}$ | $99.68_{\pm0.03}$ |
| | LLM Call (0-shot) | $99.67_{\pm0.04}$ | $100.00_{\pm0.00}$ | $99.42_{\pm0.07}$ | $99.60_{\pm0.00}$ | $99.67_{\pm0.28}$ | - | | |
| | LLM Call (3-shot) | $99.14_{\pm0.18}$ | $97.15_{\pm0.35}$ | $99.43_{\pm0.07}$ | $99.87_{\pm0.12}$ | $99.51_{\pm0.49}$ | - | | |
| Qwen3-32B | Avg-MLP | $95.88_{\pm3.31}$ | $100.00_{\pm0.00}$ | $99.99_{\pm0.01}$ | $86.87_{\pm5.22}$ | $100.00_{\pm0.00}$ | $99.97_{\pm0.01}$ | $78.67_{\pm0.67}$ | $99.93_{\pm0.02}$ |
| | Tail-MLP | $99.39_{\pm0.00}$ | $100.00_{\pm0.00}$ | $99.98_{\pm0.00}$ | $96.80_{\pm0.40}$ | $99.02_{\pm0.49}$ | $98.41_{\pm0.10}$ | $74.07_{\pm1.15}$ | $99.84_{\pm0.01}$ |
| | Avg-Linear | $98.61_{\pm1.59}$ | $100.00_{\pm0.00}$ | $99.99_{\pm0.02}$ | $87.53_{\pm6.94}$ | $100.00_{\pm0.00}$ | $99.97_{\pm0.01}$ | $78.21_{\pm0.19}$ | $99.94_{\pm0.01}$ |
| | NormStat:KL | $97.93_{\pm0.04}$ | $24.59_{\pm0.35}$ | $99.78_{\pm0.06}$ | $97.93_{\pm0.12}$ | $100.00_{\pm0.00}$ | $57.03_{\pm0.43}$ | $35.14_{\pm0.25}$ | $89.62_{\pm0.08}$ |
| | VecStat:KL | $100.00_{\pm0.00}$ | $99.39_{\pm0.00}$ | $99.98_{\pm0.03}$ | $96.60_{\pm0.00}$ | $100.00_{\pm0.00}$ | $99.53_{\pm0.07}$ | $53.16_{\pm0.52}$ | $98.74_{\pm0.05}$ |
| | VecStat:Cos | $100.00_{\pm0.00}$ | $98.17_{\pm0.00}$ | $99.98_{\pm0.03}$ | $96.80_{\pm0.35}$ | $100.00_{\pm0.00}$ | $99.61_{\pm0.06}$ | $56.70_{\pm0.88}$ | $99.58_{\pm0.02}$ |
| | LLM Call (0-shot) | $86.91_{\pm0.29}$ | $100.00_{\pm0.00}$ | $98.69_{\pm0.06}$ | $96.27_{\pm0.42}$ | $100.00_{\pm0.00}$ | - | | |
| | LLM Call (3-shot) | $97.80_{\pm0.20}$ | $100.00_{\pm0.00}$ | $98.87_{\pm0.05}$ | $99.80_{\pm0.20}$ | $100.00_{\pm0.00}$ | - | | |

**Level-1 classification: All methods perform well.** Across easy tasks, all methods achieve strong in-distribution performance. The effectiveness of NormStat, despite using only radial statistics, demonstrates itself as a computational- and memory-efficient approach when the classes are different enough. Surprisingly, NormStat outperforms all other methods in some cases (e.g. Qwen3-32B evaluated on math500). VecStat achieves marginally higher accuracies than NormStat in most cases. Training-based methods perform comparably to VecStat, with no approach consistently dominating across datasets. Avg-Linear performs comparably to Avg-MLP but with slightly lower accuracy, consistent with the reduced expressiveness of a linear layer versus two-layer architecture. Direct LLM inference shows improved classification accuracy with larger models and in-context learning, though at the cost of substantially increased prompt length and careful prompt design to achieve a reasonable performance. Notably, Llama-3.2-1B models fail to produce valid outputs even with in-context examples, hence their results are omitted. Out-of-distribution generalization varies substantially, as evidenced by the performance difference between GSM8K and MATH500 for mathematical tasks, suggesting that different methods might capture distinct aspects of domain characteristics.

**Level-2 classification: Training-based methods win, VecStat remains competitive.** Fine-grained classification within domains reveals clear performance stratification across methods. For programming language identification, VecStat maintains near-perfect accuracy while NormStat shows substantial degradation, aligning with our theoretical prediction that directional information becomes critical for within-domain discrimination. As detailed in Table 3, this performance gap remains consistent across all nine programming languages. Mathematical subfield classification proves most challenging for training-free methods, with both NormStat and VecStat falling short of training-based approaches. This gap suggests that effective discrimination among mathematical topics requires non-linear transformations better captured through supervised training signals. In contrast, natural language identification demonstrates strong performance across all methods, with VecStat achieving near-perfect accuracy, likely due to distinct linguistic features being well-separated in the LLM's hidden representation space.

**Distance metric comparison** The cosine distance variant of VecStat consistently outperforms its KL divergence counterpart, particularly in Level-2 tasks. This may suggest that angular separation between prompt and baseline statistics offers a more robust measure that captures directionality.

Table 3: Level-2 programming language classification results for Qwen3-8B and Qwen3-32B. Values represent per-class accuracy across nine programming languages from the Magicoder.

| Model | Method | cpp | csharp | java | php | python | rust | shell | swift | typescript |
|---|---|---|---|---|---|---|---|---|---|---|
| Qwen3-8B | Avg-MLP | $99.95_{\pm0.05}$ | $100.00_{\pm0.00}$ | $100.00_{\pm0.00}$ | $99.96_{\pm0.06}$ | $99.91_{\pm0.01}$ | $100.00_{\pm0.00}$ | $100.00_{\pm0.00}$ | $100.00_{\pm0.00}$ | $99.94_{\pm0.03}$ |
| | Tail-MLP | $99.61_{\pm0.16}$ | $99.69_{\pm0.18}$ | $99.47_{\pm0.12}$ | $100.00_{\pm0.00}$ | $99.39_{\pm0.22}$ | $99.53_{\pm0.10}$ | $99.92_{\pm0.14}$ | $99.75_{\pm0.07}$ | $99.61_{\pm0.10}$ |
| | Avg-Linear | $99.97_{\pm0.03}$ | $100.00_{\pm0.00}$ | $100.00_{\pm0.00}$ | $99.96_{\pm0.06}$ | $99.91_{\pm0.03}$ | $100.00_{\pm0.00}$ | $100.00_{\pm0.00}$ | $100.00_{\pm0.00}$ | $99.93_{\pm0.03}$ |
| | NormStat:KL | $47.37_{\pm0.94}$ | $38.26_{\pm1.28}$ | $27.59_{\pm0.65}$ | $60.14_{\pm1.27}$ | $67.73_{\pm0.54}$ | $63.05_{\pm0.43}$ | $91.51_{\pm0.60}$ | $62.54_{\pm1.25}$ | $49.28_{\pm0.84}$ |
| | VecStat:KL | $99.09_{\pm0.11}$ | $98.58_{\pm0.42}$ | $97.78_{\pm0.09}$ | $99.78_{\pm0.22}$ | $99.01_{\pm0.04}$ | $99.81_{\pm0.00}$ | $99.76_{\pm0.00}$ | $99.91_{\pm0.09}$ | $99.38_{\pm0.20}$ |
| | VecStat:Cos | $98.90_{\pm0.24}$ | $98.96_{\pm0.38}$ | $98.49_{\pm0.03}$ | $99.85_{\pm0.13}$ | $99.13_{\pm0.12}$ | $99.55_{\pm0.07}$ | $100.00_{\pm0.00}$ | $99.81_{\pm0.12}$ | $99.35_{\pm0.19}$ |
| Qwen3-32B | Avg-MLP | $99.91_{\pm0.03}$ | $99.98_{\pm0.04}$ | $100.00_{\pm0.00}$ | $99.96_{\pm0.06}$ | $99.91_{\pm0.03}$ | $99.97_{\pm0.06}$ | $100.00_{\pm0.00}$ | $100.00_{\pm0.00}$ | $99.96_{\pm0.04}$ |
| | Tail-MLP | $98.01_{\pm0.41}$ | $97.27_{\pm0.95}$ | $97.99_{\pm0.53}$ | $98.77_{\pm0.22}$ | $98.26_{\pm0.21}$ | $99.21_{\pm0.06}$ | $98.97_{\pm0.60}$ | $98.71_{\pm0.48}$ | $98.50_{\pm0.07}$ |
| | Avg-Linear | $99.91_{\pm0.03}$ | $99.98_{\pm0.04}$ | $100.00_{\pm0.00}$ | $99.96_{\pm0.06}$ | $99.92_{\pm0.00}$ | $99.97_{\pm0.06}$ | $100.00_{\pm0.00}$ | $100.00_{\pm0.00}$ | $99.96_{\pm0.04}$ |
| | NormStat:KL | $53.57_{\pm0.94}$ | $36.94_{\pm1.38}$ | $25.47_{\pm1.43}$ | $59.47_{\pm0.57}$ | $61.66_{\pm1.40}$ | $67.63_{\pm0.54}$ | $89.29_{\pm0.71}$ | $69.35_{\pm1.19}$ | $49.84_{\pm1.08}$ |
| | VecStat:KL | $99.01_{\pm0.11}$ | $99.35_{\pm0.34}$ | $98.72_{\pm0.03}$ | $99.89_{\pm0.11}$ | $99.29_{\pm0.06}$ | $99.90_{\pm0.05}$ | $100.00_{\pm0.00}$ | $99.85_{\pm0.09}$ | $99.75_{\pm0.16}$ |
| | VecStat:Cos | $99.18_{\pm0.14}$ | $99.53_{\pm0.19}$ | $99.01_{\pm0.10}$ | $99.89_{\pm0.11}$ | $99.50_{\pm0.04}$ | $99.81_{\pm0.13}$ | $100.00_{\pm0.00}$ | $99.87_{\pm0.12}$ | $99.72_{\pm0.09}$ |



| | Math | Text | Code |
|---|---|---|---|

(a) NormStat    (b) VecStat    (c) Tail-MLP    (d) Avg-MLP

Figure 4: Uncertainty quantification results on a mixed-intent prompt on Qwen-1.7B-Base.

## 3.3 CASE STUDY ON AMBIGUOUS PROMPTS

We examine classification uncertainty of different methods when encountering ambiguous prompts with unclear intents. We concatenate programming and mathematical content into a single prompt and analyze the resulting prediction distributions across all three level-1 categories. For NormStat and VecStat, we form probabilities via a softmax over negative distances $p_i \propto \exp(-d_i/\tau)$, and rescale distances with a problem-scale constant $\tau = O(\text{distance})$ before the softmax (in our experiments, $\tau = 10$). This temperature-like scaling yields meaningful confidence estimates with essentially zero extra cost. In contrast, the training-based approaches (Avg-MLP and Tail-MLP) take the softmax of MLP logits; these probabilities are not calibrated by default, and obtaining reliable uncertainty quantification requires an explicit post-hoc calibration step (e.g., temperature scaling learned on a validation set) (Guo et al., 2017), which we do not apply here. Both VecStat and NormStat (Fig. 4(a)–(b)) produce well-calibrated distributions that reflect the prompt's mixed nature—assigning substantial mass to both math and code while down-weighting text. In contrast, the training-based methods (Fig. 4(c)–(d)) are overconfident, placing essentially all mass on a single class despite mixed content. This failure stems from discriminative training on clean data yielding overly sharp decision boundaries that are poorly calibrated for out-of-distribution, noisy inputs. While such predictions can score well on clean test sets, they produce misleading uncertainty in deployment scenarios where mixed-intent prompts naturally occur (see Appendix D.3).

## 3.4 ROBUSTNESS TO ADVERSARIAL ATTACK

We evaluate robustness under adversarial rephrasing on MATH500, with all methods calibrated or trained only on clean data. (See Appendix D.7 for full results). As shown in Table 4, every approach suffers a performance drop on adversarial prompts, confirming that this attack meaningfully disrupts intent cues. Among training-based methods, Tail-MLP is notably robust, retaining most of its accuracy. Among training-free approaches, NormStat is the most resilient, showing only a modest drop. This aligns with the method design: NormStat relies on radial statistics that are relatively invariant to directional perturbations induced by adversarial paraphrasing.

Table 4: Level-1 accuracy on clean and adversarial MATH500 prompts for Llama-3.2-1B-Instruct.

| Method | math500 | adv_math500 |
|---|---|---|
| Tail-MLP | $94.53_{\pm0.42}$ | $82.33_{\pm3.06}$ |
| NormStat:KL | $96.20_{\pm0.00}$ | $92.40_{\pm0.00}$ |

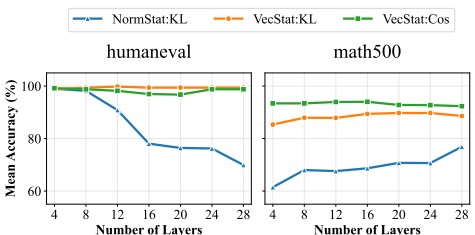 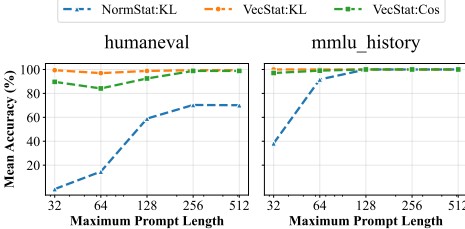

Figure 5: Effect of the number of layers on level-1 classification accuracy for Qwen3-1.7B.

Figure 6: Effect of the maximum prompt length on level-1 classification for Qwen3-1.7B.

### 3.5 EFFECT OF NUMBER OF PROBED LAYERS AND PROMPT LENGTH

**Number of probed layers** In Fig. 5, VecStat demonstrates a robust performance regardless of layer count, indicating that early layers capture sufficient statistical information for intent classification. In contrast, NormStat exhibits dataset-dependent behavior: performance degrades with additional layers on humaneval but improves on math500, though both achieve competitive accuracy using only the first 12 layers out of 28. These findings imply that routing can be performed without completing a full forward pass, substantially saving computation costs. See Appendix D.4 for additional results.

**Maximum prompt length** In Fig. 6, VecStat demonstrates remarkable stability, maintaining near-optimal accuracy across all sequence lengths from 32 to 512. This robustness indicates that coordinate-wise statistics capture sufficient discriminative information even from truncated prompts, enabling potential deployment optimizations through reduced sequence lengths. NormStat is more sensitive to prompt length, with accuracy improving substantially as length increases and plateauing at approximately 128 tokens. More detailed results are presented in Appendix D.6.

### 3.6 CALIBRATION ANALYSIS

To validate Theorem 2, we compare the empirical and theoretical convergence rates of NormStat and VecStat on the MagiCoder dataset and present the result in Fig. 7. We vary the calibration sample size from 512 to 32768 and conduct multiple runs with different seeds. Our results demonstrate strong empirical match with the theoretical bounds. Both methods show $\tilde{O}(N^{-0.5})$ rate for the mean error, following the predicted theoretical curves. Notably, NormStat attains much lower absolute errors, which is consistent with its dimension-free bound, whereas VecStat sits higher due to its dimension-dependent bounds. Calibration results for other LLMs are deferred to Appendix D.5.

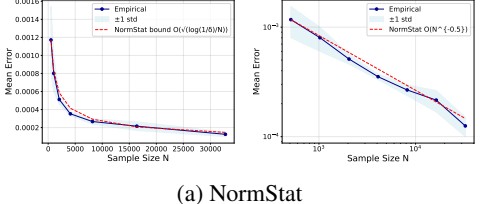 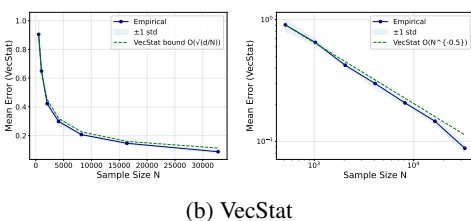

(a) NormStat

(b) VecStat

Figure 7: Calibration convergence analysis for Qwen3-8B on the MagiCoder dataset. Each subplot shows both linear and log-log scales comparing empirical results with theoretical bounds.

## 4 CONCLUSION

We conducted a comprehensive comparative study of training-free vs training-based intent classifiers and showed that both approach excel at easy classification tasks (e.g. math vs code), while training-free methods (NormStat and VecStat) provide fast, uncertainty-aware routing with minimal overhead. Training-free methods are more robust to ambiguous prompts while training-based classifiers (Avg-MLP, Tail-MLP) are better on more fine-grained classification tasks.

## STATEMENT OF AUTHORS

### ETHICS STATEMENT

We adhere to the ICLR Code of Ethics. Our study does not involve human subjects, personally identifiable information, or sensitive attributes; all datasets are publicly available under permissive licenses, and we follow their terms of use. We neither collect new data nor perform any form of user profiling, re-identification, or demographic inference. We evaluate and report results using standard, publicly accepted protocols, and we avoid releasing models or artifacts that are reasonably likely to enable harmful applications. We disclose computing resources and consider environmental impact in our experiments; no conflicts of interest or external sponsorships influenced the work. The authors take full responsibility for the integrity and accuracy of the content.

### REPRODUCIBILITY STATEMENT

We provide runnable code in the supplementary materials. The full codebase will be released as open-source after the review process. All experimental settings—including dataset specifications, preprocessing steps, training/evaluation pipelines, and exact hyperparameters—are documented in the appendix for complete reproducibility.

### THE USE OF LARGE LANGUAGE MODELS

We used LLMs solely as general-purpose assistive tools. For writing, we employed OpenAI's GPT-5 to polish language in sentence level—improving clarity, grammar, and style—without generating scientific claims, interpreting results, or drafting sections de novo. For coding, we used the Cursor IDE's built-in autocomplete to suggest boilerplate and minor edits; all coding/writing logic was authored, reviewed, and verified by the authors. The research ideas, experimental design, and overall manuscript structure were conceived and developed by the authors without any LLM involvement. The authors take full responsibility for the content.

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

## A ADDITIONAL THEORETICAL ANALYSIS

### A.1 TWO ENDPOINTS ON THE COMPRESSION LADDER

For each class $k \in [m]$, per-class baselines are computed at the same module $W_\ell$ on calibration data. We compare the two methods in terms of FLOPs and memory.

**VecStat.** **Method 1:** Compute $(S_{\text{vec}}, Q_{\text{vec}})$ as in (1). With per-class parameters $(\mu_k, \Sigma_k = \text{Diag}(\sigma_{k,1}^2, \ldots, \sigma_{k,d}^2))$. The log-likelihood ratio (LLR) between classes $i$ and $j$ is

$$\log \frac{p_i(Y)}{p_j(Y)} = -\frac{T}{2} \sum_{m=1}^{d} \left[ \log \frac{\sigma_{i,m}^2}{\sigma_{j,m}^2} + \frac{Q_{\text{vec},m} - 2\mu_{i,m} S_{\text{vec},m} + \mu_{i,m}^2}{\sigma_{i,m}^2} - \frac{Q_{\text{vec},m} - 2\mu_{j,m} S_{\text{vec},m} + \mu_{j,m}^2}{\sigma_{j,m}^2} \right],$$
(4)

which is equivalently the average of coordinate-wise Gaussian KLs (since $\Sigma_k$ is diagonal):

$$\sum_{m=1}^{d} \text{KL}\big(\mathcal{N}(\mu_{i,m}, \sigma_{i,m}^2) \,\|\, \mathcal{N}(\mu_{j,m}, \sigma_{j,m}^2)\big) = \frac{1}{2} \sum_{m=1}^{d} \left[ \log \frac{\sigma_{j,m}^2}{\sigma_{i,m}^2} + \frac{\sigma_{i,m}^2 + (\mu_{i,m} - \mu_{j,m})^2}{\sigma_{j,m}^2} - 1 \right].$$
(5)

**Method 2:** Using $S_{\text{vec}}$, classify via

$$\cos_k(S_{\text{vec}}, \mu_k) = \frac{\langle S_{\text{vec}}, \mu_k \rangle}{\|S_{\text{vec}}\| \|\mu_k\|}, \qquad \hat{b} = \arg \max_{k \in [m]} \cos_k(S_{\text{vec}}, \mu_k).$$

**Costs:** *Per-token compute:* $\Theta(d)$. *Prompt-state:* $O(d)$. *Baseline storage:* $O(md)$ *numbers. (If only cosine scoring is used,* $Q_{\text{vec}}$ *need not be stored.)*

**NormStat.** **Method:** Compute $(S_{\text{norm}}, Q_{\text{norm}})$ as in (2), then compare $(S_{\text{norm}}, Q_{\text{norm}})$ to per-class baselines $(\mu_{x,k}, \sigma_{x,k}^2)$ via a 1D Gaussian KL:

$$\text{KL}\big(\mathcal{N}(\mu_{x,i}, \sigma_{x,i}^2) \| \mathcal{N}(\mu_{x,j}, \sigma_{x,j}^2)\big) = \frac{1}{2} \left[ \log \frac{\sigma_{x,j}^2}{\sigma_{x,i}^2} + \frac{\sigma_{x,i}^2 + (\mu_{x,i} - \mu_{x,j})^2}{\sigma_{x,j}^2} - 1 \right]. \tag{6}$$

**Cost.** *Per-token compute:* $\Theta(d)$. *Prompt-state:* $O(1)$. *Baseline storage:* $O(m)$ *scalars.*

### A.2 SUFFICIENCY

We establish the minimal sufficiency of $(S_{\text{vec}}, Q_{\text{vec}})$ for the diagonal–Gaussian model, which is a classical result, see (Lehmann & Casella, 1998; Lehmann & Scheffé, 2011). Intuitively, a sufficient statistic is a lossless compression for inference about the unknown class/parameters: once $(S_{\text{vec}}, Q_{\text{vec}})$ is known, the raw sample $Y$ contains no further information. Minimal sufficiency means no additional compression is possible without losing information—every other sufficient statistic is a measurable function of $(S_{\text{vec}}, Q_{\text{vec}})$.

**Lemma 1.** *Under* (3) *with diagonal* $\Sigma_k$, *the pair* $(S_{\text{vec}}, Q_{\text{vec}})$ *is a* minimal sufficient statistic *for the family* $\{\mathcal{N}(\mu_k, \Sigma_k)^{\otimes T}\}$, *and the class LLR* (4) *depends on the data only through* $(S_{\text{vec}}, Q_{\text{vec}})$.

The proof is provided in Appendix B.

## B PROOFS

### B.1 PROOF OF LEMMA 1

*Proof of Lemma 1.* Let $Y := (y_1, \ldots, y_T) \in \mathbb{R}^{T \times d}$. For a class $k$, let

$$\theta_k = (\mu_k, \sigma_{k,1}^2, \ldots, \sigma_{k,d}^2), \qquad \Sigma_k = \text{Diag}(\sigma_{k,1}^2, \ldots, \sigma_{k,d}^2).$$

Define the token-wise *sums*

$$S := \sum_{t=1}^{T} y_t \in \mathbb{R}^d, \qquad Q := \sum_{t=1}^{T} (y_t \odot y_t) \in \mathbb{R}^d,$$

and write $S_m := \sum_{t=1}^{T} y_{t,m}$, $Q_m := \sum_{t=1}^{T} y_{t,m}^2$ for coordinates $m = 1, \ldots, d$. These relate to the averaged statistics in (1) as follows:

$$S = TS_{\text{vec}}, \qquad Q = (T-1)Q_{\text{vec}} + T(S_{\text{vec}} \odot S_{\text{vec}})$$

Since (minimal) sufficiency is invariant under invertible reparameterizations of the statistic, we may work with $(S, Q)$ and translate back to $(S_{\text{vec}}, Q_{\text{vec}})$ via the identities above.

**Sufficiency.** By (3), $y_t$ are i.i.d. with density

$$p_{\theta_k}(y) = \frac{1}{(2\pi)^{d/2} \prod_{m=1}^{d} \sigma_{k,m}} \exp\left( -\frac{1}{2} \sum_{m=1}^{d} \frac{(y_m - \mu_{k,m})^2}{\sigma_{k,m}^2} \right).$$

Hence the joint density of $Y$ under class $k$ is

$$p_{\theta_k}(Y) = \frac{1}{(2\pi)^{Td/2} \prod_{m=1}^{d} \sigma_{k,m}^T} \exp\left( -\frac{1}{2} \sum_{t=1}^{T} \sum_{m=1}^{d} \frac{(y_{t,m} - \mu_{k,m})^2}{\sigma_{k,m}^2} \right)$$

$$= \underbrace{\frac{1}{(2\pi)^{Td/2}}}_{:=h(Y)} \cdot \underbrace{\frac{1}{\prod_{m=1}^{d} \sigma_{k,m}^T} \exp\left( -\frac{1}{2} \sum_{m=1}^{d} \frac{Q_m - 2\mu_{k,m}S_m + T\mu_{k,m}^2}{\sigma_{k,m}^2} \right)}_{:=g_{\theta_k}(S,Q)}.$$

Thus $p_{\theta_k}(Y) = h(Y) \, g_{\theta_k}(S, Q)$. By the Neyman–Fisher factorization theorem, $(S, Q)$ is sufficient for $\theta_k$, and hence $(S_{\text{vec}}, Q_{\text{vec}})$ is sufficient by the invertible mapping above.

**Minimality.** Using the Lehmann–Scheffé characterization: a statistic $T(Y)$ is minimal sufficient iff for any $Y, Y'$ the likelihood ratio $p_\theta(Y)/p_\theta(Y')$ is free of $\theta$ if and only if $T(Y) = T(Y')$. For our family,

$$\frac{p_\theta(Y)}{p_\theta(Y')} = \exp\left( -\frac{1}{2} \sum_{m=1}^{d} \frac{(Q_m - Q_m') - 2\mu_m(S_m - S_m')}{\sigma_m^2} \right).$$

If $(S, Q) = (S', Q')$ then this ratio equals 1, hence is parameter–free. Conversely, if for some $m$ either $S_m \neq S_m'$ or $Q_m \neq Q_m'$, the exponent depends on $\mu_m$ (when $S_m \neq S_m'$) or on $\sigma_m^2$ (when $Q_m \neq Q_m'$); thus the ratio cannot be constant in $\theta$.

Moreover, for classes $i$ and $j$, subtracting the two log-likelihoods above yields

$$\log \frac{p_i(Y)}{p_j(Y)} = -\frac{1}{2} \sum_{m=1}^{d} \left( T \log \frac{\sigma_{i,m}^2}{\sigma_{j,m}^2} + \frac{Q_m - 2\mu_{i,m}S_m + T\mu_{i,m}^2}{\sigma_{i,m}^2} - \frac{Q_m - 2\mu_{j,m}S_m + T\mu_{j,m}^2}{\sigma_{j,m}^2} \right),$$

which is exactly (4) and depends on $Y$ only through $(S, Q)$, and equivalently only through $(S_{\text{vec}}, Q_{\text{vec}})$ via the identities at the start of the proof.

This establishes that $(S_{\text{vec}}, Q_{\text{vec}})$ is minimal sufficient and that the LLR depends on the sample only through this pair.

$\square$

### B.2 PROOF OF THEOREM 1

*Proof of Theorem 1.* **Directional regime:** Since $\|\mu_1\| = \|\mu_2\|$, there exists an orthogonal matrix $U$ with $U\mu_1 = \mu_2$. If $Y \sim \mathcal{N}(\mu_1, \sigma^2 I_d)$ then $UY \sim \mathcal{N}(\mu_2, \sigma^2 I_d)$ and $\|UY\| = \|Y\|$. Thus for each $t$, $\|y_t\| \mid k = 1$ and $\|y_t\| \mid k = 2$ have the same distribution, and by independence the vectors $(\|y_t\|)_{t=1}^{T} \mid k = 1$ and $(\|y_t\|)_{t=1}^{T} \mid k = 2$ are identically distributed. With a uniform prior, any decision rule that depends only on $\{\|y_t\|\}$ has the same acceptance probability under both classes, so its Bayes error is $1/2$.

For the log-likelihood ratio test (LRT), the log-likelihood ratio for two Gaussians with common covariance $\sigma^2 I_d$ is

$$\Lambda(y_{1:T}) = \frac{T}{\sigma^2} \left\langle S_{\text{vec}}, \mu_1 - \mu_2 \right\rangle - \frac{T}{2\sigma^2} \left( \|\mu_1\|^2 - \|\mu_2\|^2 \right).$$

With equal priors the LRT accepts $k = 1$ iff $\Lambda \geq 0$. Under $\|\mu_1\| = \|\mu_2\|$, the constant term vanishes and the decision reduces to the sign of $\langle S_{\text{vec}}, \mu_1 - \mu_2 \rangle$, i.e., to $\hat{k}$ above.

Let $u := (\mu_1 - \mu_2)/\|\mu_1 - \mu_2\|$ and $Z := \langle S_{\text{vec}}, u \rangle$. Since $S_{\text{vec}} \mid k \sim \mathcal{N}(\mu_k, \frac{\sigma^2}{T} I_d)$ and $\|u\| = 1$,

$$Z \mid k \sim \mathcal{N}\left( \langle \mu_k, u \rangle, \frac{\sigma^2}{T} \right), \quad \langle \mu_1, u \rangle = \tfrac{1}{2}\|\mu_1 - \mu_2\|, \ \ \langle \mu_2, u \rangle = -\tfrac{1}{2}\|\mu_1 - \mu_2\|.$$

Hence, by symmetry,

$$\Pr\left( \hat{k}(y_{1:T}) \neq k \right) = \Pr_{k=1} (Z < 0) = \Phi\left( -\frac{\|\mu_1 - \mu_2\|}{2\sigma} \sqrt{T} \right) \leq \exp\left( -\frac{T}{8\sigma^2} \|\mu_1 - \mu_2\|^2 \right),$$

where $\Phi$ is the standard normal CDF and the last step uses $\Phi(-x) \leq e^{-x^2/2}$ for $x \geq 0$.

**Isotropic-scale regime.** Let $\phi_d(\,\cdot\,; m, \Sigma)$ denote the $d$-variate Gaussian density. For $k \in \{1, 2\}$, the *joint* density of $y_{1:T}$ under class $k$ is $p_k(y_{1:T}) := \prod_{t=1}^{T} \phi_d\left( y_t; 0, \sigma_k^2 I_d \right)$.

With $\mu_1 = \mu_2 = 0$, $\Sigma_k = \sigma_k^2 I_d$, and $R_T = \sum_{t=1}^{T} \|y_t\|^2$, one can calculate the log-likelihood ratio

$$\log \frac{p_1(y_{1:T})}{p_2(y_{1:T})} = \sum_{t=1}^{T} \log \frac{\phi_d(y_t; 0, \sigma_1^2 I_d)}{\phi_d(y_t; 0, \sigma_2^2 I_d)} = \frac{dT}{2} \log \frac{\sigma_2^2}{\sigma_1^2} + \frac{1}{2} \left( \frac{1}{\sigma_2^2} - \frac{1}{\sigma_1^2} \right) R_T.$$

The right-hand side is an affine (hence strictly monotone when $\sigma_1 \neq \sigma_2$) function of $R_T$. By the Neyman–Pearson lemma, any Bayes–optimal test is a threshold on $R_T$, so purely radial statistics are sufficient for optimality and coordinate-wise additions cannot lower the Bayes risk. $\qquad\square$

### B.3 Proof of Theorem 2

**Theorem 3.** *Fix a class $k$. Let $y_1, \ldots, y_N \overset{i.i.d.}{\sim} \mathcal{N}(\mu_k, \Sigma_k)$ in $\mathbb{R}^d$, where $N$ is the number of calibration samples drawn for this class. Define*

$$\hat{\mu}_k := \frac{1}{N} \sum_{i=1}^{N} y_i, \quad \hat{q} := \frac{1}{N} \sum_{i=1}^{N} \|y_i\|^2, \quad \sigma_{\max}^2 := \|\Sigma_k\|_{\text{op}}.$$

*For coordinate variances, write $\sigma_{k,j}^2 := (\Sigma_k)_{jj}$ and $\hat{\sigma}_{k,j}^2 := \frac{1}{N} \sum_{i=1}^{N} (y_{i,j} - \hat{\mu}_{k,j})^2$, $j = 1, \ldots, d$. Then:*

1. ***NormStat (dimension-free).** For $q = \|y\|^2$, one has*

$$\mathbb{E}[q] = \|\mu_k\|^2 + \text{Tr}(\Sigma_k), \quad \text{Var}(q) = 2 \text{Tr}(\Sigma_k^2) + 4\mu_k^\top \Sigma_k \mu_k.$$

*By Bernstein's inequality for sub-exponential variables, for all $\delta \in (0, 1)$,*

$$|\hat{q} - q| \lesssim \sqrt{\frac{\text{Var}(q) \log(1/\delta)}{N}} \quad \text{with probability at least } 1 - \delta.$$

*Normalizing by $d$ makes the bound $O(\sqrt{\log(1/\delta)/N})$, i.e. dimension-free.*

2. ***VecStat (dimension-dependent).** With probability at least $1 - \delta$,*

$$\|\hat{\mu}_k - \mu_k\|_2 \leq C_1 \sigma_{\max} \sqrt{\frac{d + \log(1/\delta)}{N}}, \quad \max_{j} \left| \hat{\sigma}_{k,j}^2 - \sigma_{k,j}^2 \right| \leq C_2 \sigma_{\max}^2 \sqrt{\frac{\log(d/\delta)}{N}},$$

*for absolute constants $C_1, C_2$. To keep LLR plug-in error small of order $\epsilon$, one needs $N = \Omega(d/\epsilon_\mu^2)$ for mean accuracy in $\ell_2$ and $N = \Omega(\log d/\epsilon_\sigma^2)$ for variances in $\ell_\infty$.*

*Proof of Theorem 2.* **NormStat:** Denote $q_i := \|y_i\|^2$ and $Z_i := q_i - \mathbb{E}[q]$, so that $\hat{q} - \mathbb{E}[q] = \frac{1}{N}\sum_{i=1}^{N} Z_i$. For $y_i \sim \mathcal{N}(\mu_k, \Sigma_k)$, the centered quadratic form $Z_i$ obeys the Hanson–Wright tail bound: there exist absolute constants $c_1, c_2 > 0$ such that for all $t > 0$,

$$\Pr\left(|Z_i| \geq t\right) \leq 2\exp\left[-c_1 \min\left(\frac{t^2}{\mathrm{Var}(q)}, \frac{t}{B}\right)\right], \tag{7}$$

where $\mathrm{Var}(q) = 2\,\mathrm{Tr}(\Sigma_k^2) + 4\,\mu_k^\top \Sigma_k \mu_k$, $B = \|\Sigma_k\|_{\mathrm{op}} + \|\mu_k\|^2$. From (7), the $Z_i$ are i.i.d. mean-zero sub-exponential. A standard Bernstein inequality for sums of independent sub-exponential variables then yields, for some absolute $c > 0$ and all $t > 0$,

$$\Pr\left(\left|\frac{1}{N}\sum_{i=1}^{N} Z_i\right| \geq t\right) \leq 2\exp\left[-cN \min\left(\frac{t^2}{\mathrm{Var}(q)}, \frac{t}{B}\right)\right].$$

Choosing $t \lesssim \mathrm{Var}(q)/B$ and inverting the tail gives, for any $\delta \in (0,1)$,

$$|\hat{q} - \mathbb{E}[q]| \lesssim \sqrt{\frac{\mathrm{Var}(q)\log(2/\delta)}{N}} \quad \text{with probability at least } 1 - \delta.$$

Since under bounded eigenvalues $\mathrm{Var}(q) = \Theta(d)$, dividing by $d$ yields $\left|\frac{1}{d}\hat{q} - \frac{1}{d}\mathbb{E}[q]\right| \lesssim \sqrt{\frac{\log(2/\delta)}{N}}$, which is dimension-free.

**VecStat:** Let $z_i := \Sigma_k^{-1/2}(y_i - \mu_k) \sim \mathcal{N}(0, I_d)$. Then

$$\hat{\mu}_k - \mu_k = \Sigma_k^{1/2}\left(\frac{1}{N}\sum_{i=1}^{N} z_i\right) \sim \mathcal{N}\left(0, \frac{1}{N}\Sigma_k\right).$$

Hence, $\|\Sigma_k^{-1/2}(\hat{\mu}_k - \mu_k)\|_2^2 \sim \frac{1}{N}\chi_d^2$. Recall the standard Laurent-Massart inequalities: for any $x > 0$,

$$\Pr\left(\chi_d^2 - d \geq 2\sqrt{dx} + 2x\right) \leq e^{-x}, \quad \Pr\left(d - \chi_d^2 \geq 2\sqrt{dx}\right) \leq e^{-x}. \tag{8}$$

Applying (8) with $x = \log(1/\delta)$ and scaling by $1/N$ yields, with probability $\geq 1 - \delta$,

$$\|\Sigma_k^{-1/2}(\hat{\mu}_k - \mu_k)\|_2 \leq \sqrt{\frac{1}{N}\left(d + 2\sqrt{d\log(1/\delta)} + 2\log(1/\delta)\right)}.$$

Multiplying by $\|\Sigma_k^{1/2}\|_{\mathrm{op}} = \sigma_{\max}$ and using $\sqrt{a+b} \leq \sqrt{a} + \sqrt{b}$ gives

$$\|\hat{\mu}_k - \mu_k\|_2 \leq C_1 \sigma_{\max}\sqrt{\frac{d + \log(1/\delta)}{N}},$$

for an absolute constant $C_1 > 0$.

$\square$

## C   EXPERIMENT DETAILS

### C.1   DATASET COMPOSITION AND PROCESSING STRATEGIES

Table 5: Datasets composition for level-1 classification.

| Category | Calibration Data | # Calibration Samples | Classification Data | # Classification Samples |
|---|---|---|---|---|
| General Text | MMLU (European History) | 165 | MMLU (US History) | 204 |
| Math | GSM8K | 2,000 | GSM8K
MATH500 | 1,319
500 |
| Code | Magicoder | 2,000 | Magicoder
HumanEval | 5,000
164 |

We evaluate our lightweight intent classification methods across two hierarchical granularities. Our experimental protocol consists of two stages: calibration and classification. During calibration, we compute per-class baseline statistics (NormStat or VecStat) from calibration data passed through pretrained LLMs. During classification, we compute the same statistics for test prompts and assign labels based on the minimum KL divergence (or cosine distance) between the prompt's statistics and the calibrated per-class baselines.

Table 6: Datasets composition for level-2 classification.

| Task | Data Source | Classes | # Calibration Samples per Class | # Classification Samples per Class |
|------|-------------|---------|----------------------------------|-------------------------------------|
| Code | Magicoder | C++, C#, Java, PHP, Python, Rust, Shell, Swift, TypeScript | 2000 | 5000 |
| Math | Competition Math | Algebra, Counting & Probability, Geometry, Intermediate Algebra, Number Theory, Prealgebra, Precalculus | 800 | 3000 |
| Natural Language | Aya | Sinhala, Tamil, English, Moroccan Arabic, Japanese | 512 | 3000 |

**Classification granularities**   We consider the following classification granularities:

- **Level-1 Classification** evaluates coarse-grained categorization into three primary domains: general text, mathematics, and code. This level represents the typical routing scenario where prompts are directed to specialized models based on broad task categories.
- **Level-2 Classification** examines fine-grained discrimination within each domain. We evaluate three distinct tasks: (i) programming language identification across nine languages in code prompts, (ii) mathematical subfield classification across seven topics, and (iii) natural language identification across five linguistically diverse languages.

**Datasets and evaluation**   Table 5 and Table 6 present the complete dataset composition. For Level-1 classification, we calibrate using domain-representative datasets: MMLU European History for general text (165 samples), GSM8K for math (2,000 samples), and Magicoder for code (2,000 samples). Classification evaluation employs both in-distribution and out-of-distribution datasets to assess generalization. Specifically, we evaluate general text on MMLU US History, mathematics on GSM8K (in-distribution) and MATH500 (out-of-distribution), and code on Magicoder (in-distribution) and HumanEval (out-of-distribution).

For Level-2 classification, we maintain consistent calibration sizes where feasible: 2,000 samples per programming language, 800 samples per mathematical subfield, and 512 samples per natural language. Classification sets contain up to 5,000 samples per programming language and 3,000 samples per category for mathematics and natural languages, subject to dataset availability. We also provide more details on the datasets used in Appendix C.2.

## C.2   DATASETS

We employ seven benchmark datasets spanning general text, mathematics, and code domains to evaluate intent classification performance at both granularity levels.

- **General Text Datasets**. For Level-1 evaluation, we utilize MMLU (Hendrycks et al., 2021a;b), a comprehensive benchmark of multiple-choice questions across 57 subjects. We construct calibration data using the High School European History subset and evaluate on the High School US History subset, using question-choice pairs as input. For Level-2 language classification, we employ the Aya dataset (Singh et al., 2024), which contains human-annotated prompts across 65 languages. We select five linguistically diverse languages as specified in Table 6 and use the input field for classification.
- **Mathematics Datasets**. We employ three mathematics benchmarks for comprehensive evaluation. GSM8K (Cobbe et al., 2021) provides grade-school word problems requiring multi-step reasoning, from which we sample 2,000 calibration instances and use the complete test set for Level-1 classification. MATH500 (Lightman et al., 2023) serves as an out-of-distribution test set containing 500 problems from the MATH benchmark. For Level-2 domain-specific classification, Competition Math (Hendrycks et al., 2021c) provides problems from mathematics competitions spanning seven mathematical subfields including algebra, geometry, and number theory. We use the problem field as model input for Level-2 domain-specific classification.
- **Code Datasets**. Magicoder (Wei et al., 2023) forms our primary code classification resource, containing solutions across multiple programming languages. We utilize the solution field as

model input for both Level-1 general code classification and Level-2 language-specific classification tasks, focusing on 9 mainstream programming languages as detailed in Table 6. HumanEval (Chen et al., 2021) provides 164 function-level programming problems, where we use the prompt field containing function signatures and docstrings as model input, serving as an out-of-distribution test set for Level-1 classification.

All datasets are publicly available through HuggingFace Datasets (Lhoest et al., 2021). The exact sampling strategies and train-test splits follow the specifications in Table 5 and Table 6, with test samples capped at the minimum of specified counts and available data.

## C.3 SELECTED LLMs

We evaluate our approach on 7 pretrained large language models spanning 1B to 32B parameters, encompassing both base and instruction-tuned variants. This selection provides comprehensive coverage across model scales and training stages. We consider the following two LLM families:

- **Qwen family** (Yang et al., 2025): We evaluate four models from the Qwen3 series. The instruction-tuned variants include Qwen3-1.7B (28 layers), Qwen3-4B (36 layers), Qwen3-8B (36 layers), and Qwen3-32B (64 layers), each post-trained with supervised fine-tuning and reinforcement learning from human feedback (RLHF). Additionally, we include Qwen3-1.7B-Base to assess performance on pretrained models without alignment. For all Qwen3 evaluations, we switch on non-thinking mode to ensure consistent comparison across models.
- **Llama family** (Dubey et al., 2024): We evaluate Llama-3.2-1B (16 layers) and its instruction-tuned counterpart Llama-3.2-1B-Instruct. The instruction-tuned variant underwent supervised fine-tuning and RLHF to better align with human preferences.

This benchmark model selection enables systematic evaluation across three critical dimensions: model scale (from 1B to 32B parameters), training paradigm (pretrained-only versus post-trained), and architectural diversity (Qwen and Llama families). The substantial range in model sizes—spanning over an order of magnitude in parameters—allows us to rigorously test whether our method can effectively operate across vastly different computational scales and model capacities. The comparison between base and aligned models reveals how post-training procedures affect our method's performance, demonstrating whether it remains equally effective for both pretrained and instruction-tuned models. The prompt used for direct LLM inference under level-1 setting is in Table 7.

## C.4 MORE IMPLEMENTATION DETAILS

To stabilize training, we normalize input features and weight matrices through the following process:

1. Input Normalization: The input tensor $z$ is normalized to unit norm for stability:
$$z_{\text{normalized}} = \frac{z}{\|z\|_2 + \epsilon}.$$

2. Weight Normalization: The weight matrix $W$ is normalized using its Frobenius norm:
$$W_{\text{normalized}} = \frac{W}{\sqrt{\text{mean}(W^2) + \epsilon}}.$$

3. Activation Computation: The normalized input is multiplied by the normalized weight matrix:
$$Wz = z_{\text{normalized}} \cdot W_{\text{normalized}}^T.$$

The statistics for NormStat and VecStat are computed from $Wz$. Note that this is likely suboptimal; in a production-scale implementation we should read $Wz$ directly from the module's output rather than recomputing it.

## C.5 PSEUDO-ALGORITHM

Our task inference approach follows the following steps:

Table 7: Prompt template for direct LLM inference under level-1 intent classification setting. The template includes placeholders {examples_block} for optional few-shot examples and {user_text} for the input to be classified. The few-shot examples are taken from the calibration data.

---

**LLM Prompt Template for Intent Classification**

---

You are a classifier.
Your task is to look at the user's input text and decide which of these three categories it belongs to:

1. General text – natural language content like sentences, questions, explanations, stories, or instructions that are not primarily mathematics or code.
2. Math – text that is primarily mathematical expressions, equations, formulas, or word problems where the main focus is on mathematics.
3. Code – text that is primarily programming code or pseudocode (any programming language, including configuration snippets or shell commands).

Output rules:

- If the input is general text, output: A
- If the input is math, output: B
- If the input is code, output: C

Important Notes:

- Output only a single letter: A, B, or C.
- Do not output anything else (no explanation, no punctuation, no spaces).

Below are some classification examples for this task:
**{examples_block}**

Now classify the following input accordingly and output just one letter.

Input:
**{user_text}**
Output:

---

**Algorithm 1** Intent Classification with NormStat or VecStat

---

**Require:** Input prompt $x$, baseline scores $\{S_1, S_2, \ldots, S_m\}$ for tasks $C_1, C_2, \ldots, C_m$, distance function "dist" (NormStat: KL; VecStat: KL or cosine similarity)
**Ensure:** Predicted task $T_{\text{pred}}$ and confidence scores
1: Compute statistic scores $S_x$ from input $x$
2: **for** each task $C_i$ **do**
3:     Compute distance $d_i = \text{dist}(S_x, S_i)$
4: **end for**
5: $C_{\text{pred}} = \arg\min_{C_i} d_i$
6: Compute probabilities via softmax: $p_i = \frac{\exp(-(d_i - \bar{d})/\tau)}{\sum_j \exp(-(d_j - \bar{d})/\tau)}$ (used for uncertainty quantification, $\bar{d}$ is the average, $\tau$ is the temperature)
7: **return** $C_{\text{pred}}, \{p_1, p_2, \ldots, p_m\}$

---

C.6    HARDWARE AND SOFTWARE ENVIRONMENT

We conducted experiments on two computational platforms based on model scales. For models up to 4B parameters, we utilized an NVIDIA L40S GPU with 48GB of memory. For larger models (Qwen3-8B and Qwen3-32B), experiments were performed on an NVIDIA Grace Hopper GH200 superchip, featuring a Grace ARM 72-core CPU with 120GB RAM and a NVIDIA H100 GPU with

96GB of memory. All experiments are implemented using Python 3.12.0 and PyTorch 2.7.0 with CUDA 12.6.

# D  ADDITIONAL EXPERIMENTAL RESULTS

## D.1  ADDITIONAL RESULTS FOR LEVEL-1 CLASSIFICATION.

Please refer to Table 8.

Table 8: Level-1 classification results for all seven LLMs.

| Model | Method | gsm8k | humaneval | magicoder | math500 | mmlu_history |
|---|---|---|---|---|---|---|
| Qwen3-1.7B | Avg-MLP | $99.97_{\pm0.04}$ | $99.80_{\pm0.35}$ | $99.99_{\pm0.02}$ | $78.33_{\pm1.67}$ | $100.00_{\pm0.00}$ |
| | Tail-MLP | $100.00_{\pm0.00}$ | $100.00_{\pm0.00}$ | $99.97_{\pm0.02}$ | $99.00_{\pm0.00}$ | $100.00_{\pm0.00}$ |
| | Avg-Linear | $100.00_{\pm0.00}$ | $99.80_{\pm0.35}$ | $100.00_{\pm0.00}$ | $69.20_{\pm9.18}$ | $100.00_{\pm0.00}$ |
| | NormStat:KL | $97.35_{\pm0.00}$ | $70.12_{\pm0.61}$ | $99.27_{\pm0.11}$ | $76.60_{\pm0.20}$ | $100.00_{\pm0.00}$ |
| | VecStat:KL | $100.00_{\pm0.00}$ | $99.39_{\pm0.00}$ | $99.97_{\pm0.03}$ | $88.33_{\pm0.30}$ | $100.00_{\pm0.00}$ |
| | VecStat:Cos | $100.00_{\pm0.00}$ | $98.78_{\pm0.00}$ | $99.97_{\pm0.03}$ | $92.26_{\pm0.11}$ | $100.00_{\pm0.00}$ |
| | LLM Call (0-shot) | $51.18_{\pm0.30}$ | $99.39_{\pm0.61}$ | $97.38_{\pm0.11}$ | $80.87_{\pm0.61}$ | $82.19_{\pm1.02}$ |
| | LLM Call (3-shot) | $99.49_{\pm0.09}$ | $78.05_{\pm1.61}$ | $99.26_{\pm0.11}$ | $99.47_{\pm0.12}$ | $73.37_{\pm1.72}$ |
| Qwen3-1.7B-Base | Avg-MLP | $78.09_{\pm31.54}$ | $99.80_{\pm0.35}$ | $100.00_{\pm0.00}$ | $40.60_{\pm19.91}$ | $100.00_{\pm0.00}$ |
| | Tail-MLP | $100.00_{\pm0.00}$ | $84.55_{\pm26.76}$ | $99.87_{\pm0.18}$ | $99.00_{\pm0.00}$ | $100.00_{\pm0.00}$ |
| | Avg-Linear | $79.76_{\pm0.53}$ | $99.59_{\pm0.70}$ | $100.00_{\pm0.00}$ | $30.73_{\pm4.28}$ | $100.00_{\pm0.00}$ |
| | NormStat:KL | $79.71_{\pm0.29}$ | $82.93_{\pm0.00}$ | $99.71_{\pm0.03}$ | $88.47_{\pm0.12}$ | $100.00_{\pm0.00}$ |
| | VecStat:KL | $40.46_{\pm0.10}$ | $100.00_{\pm0.00}$ | $100.00_{\pm0.0}$ | $49.06_{\pm0.95}$ | $100.00_{\pm0.00}$ |
| | VecStat:Cos | $99.84_{\pm0.00}$ | $99.39_{\pm0.00}$ | $99.98_{\pm0.02}$ | $92.30_{\pm0.39}$ | $100.00_{\pm0.00}$ |
| | LLM Call (0-shot) | $90.60_{\pm0.00}$ | $0.00_{\pm0.00}$ | $81.03_{\pm0.39}$ | $59.40_{\pm0.00}$ | $4.41_{\pm0.00}$ |
| | LLM Call (3-shot) | $64.54_{\pm0.88}$ | $33.54_{\pm3.23}$ | $85.71_{\pm0.26}$ | $80.87_{\pm1.10}$ | $58.17_{\pm3.93}$ |
| Llama-3.2-1B | Avg-MLP | $100.00_{\pm0.00}$ | $100.00_{\pm0.00}$ | $99.99_{\pm0.01}$ | $64.33_{\pm8.33}$ | $99.84_{\pm0.28}$ |
| | Tail-MLP | $99.95_{\pm0.04}$ | $100.00_{\pm0.00}$ | $99.97_{\pm0.03}$ | $98.93_{\pm0.42}$ | $99.35_{\pm1.13}$ |
| | Avg-Linear | $100.00_{\pm0.00}$ | $99.80_{\pm0.35}$ | $99.99_{\pm0.01}$ | $71.47_{\pm7.78}$ | $98.69_{\pm1.86}$ |
| | NormStat:KL | $99.49_{\pm0.09}$ | $90.85_{\pm0.00}$ | $96.39_{\pm0.10}$ | $83.40_{\pm0.00}$ | $92.48_{\pm0.57}$ |
| | VecStat:KL | $100.00_{\pm0.00}$ | $99.39_{\pm0.00}$ | $99.98_{\pm0.02}$ | $78.80_{\pm0.12}$ | $100.00_{\pm0.00}$ |
| | VecStat:Cos | $100.00_{\pm0.00}$ | $99.39_{\pm0.00}$ | $99.97_{\pm0.02}$ | $77.60_{\pm0.20}$ | $100.00_{\pm0.00}$ |
| Llama-3.2-1B-Instruct | Avg-MLP | $100.00_{\pm0.00}$ | $99.59_{\pm0.35}$ | $99.99_{\pm0.02}$ | $72.40_{\pm2.91}$ | $100.00_{\pm0.00}$ |
| | Tail-MLP | $100.00_{\pm0.00}$ | $100.00_{\pm0.00}$ | $99.89_{\pm0.05}$ | $94.53_{\pm0.42}$ | $100.00_{\pm0.00}$ |
| | Avg-Linear | $100.00_{\pm0.00}$ | $99.80_{\pm0.35}$ | $100.00_{\pm0.00}$ | $66.80_{\pm7.75}$ | $100.00_{\pm0.00}$ |
| | NormStat:KL | $100.00_{\pm0.00}$ | $65.24_{\pm0.00}$ | $97.21_{\pm0.21}$ | $96.20_{\pm0.00}$ | $98.53_{\pm0.00}$ |
| | VecStat:KL | $100.00_{\pm0.00}$ | $99.39_{\pm0.00}$ | $99.96_{\pm0.01}$ | $87.87_{\pm0.11}$ | $100.00_{\pm0.00}$ |
| | VecStat:Cos | $100.00_{\pm0.00}$ | $99.39_{\pm0.00}$ | $99.97_{\pm0.03}$ | $85.60_{\pm0.00}$ | $100.00_{\pm0.00}$ |
| Qwen3-4B | Avg-MLP | $99.95_{\pm0.09}$ | $100.00_{\pm0.00}$ | $99.99_{\pm0.01}$ | $79.27_{\pm4.39}$ | $100.00_{\pm0.00}$ |
| | Tail-MLP | $100.00_{\pm0.00}$ | $100.00_{\pm0.00}$ | $99.96_{\pm0.02}$ | $92.27_{\pm1.17}$ | $100.00_{\pm0.00}$ |
| | Avg-Linear | $99.90_{\pm0.04}$ | $100.00_{\pm0.00}$ | $99.99_{\pm0.01}$ | $68.87_{\pm18.85}$ | $100.00_{\pm0.00}$ |
| | NormStat:KL | $96.36_{\pm0.00}$ | $35.57_{\pm0.35}$ | $99.84_{\pm0.03}$ | $89.40_{\pm0.00}$ | $99.84_{\pm0.28}$ |
| | VecStat:KL | $99.97_{\pm0.04}$ | $100.00_{\pm0.00}$ | $99.99_{\pm0.02}$ | $91.73_{\pm0.23}$ | $100.00_{\pm0.00}$ |
| | VecStat:Cos | $100.00_{\pm0.00}$ | $96.34_{\pm0.00}$ | $99.97_{\pm0.03}$ | $93.40_{\pm0.20}$ | $100.00_{\pm0.00}$ |
| | LLM Call (0-shot) | $91.84_{\pm0.16}$ | $100.00_{\pm0.00}$ | $99.47_{\pm0.06}$ | $98.33_{\pm0.12}$ | $95.42_{\pm0.28}$ |
| | LLM Call (3-shot) | $97.60_{\pm0.31}$ | $91.06_{\pm1.27}$ | $99.03_{\pm0.15}$ | $99.87_{\pm0.23}$ | $96.41_{\pm1.02}$ |
| Qwen3-8B | Avg-MLP | $99.42_{\pm0.74}$ | $99.59_{\pm0.35}$ | $99.99_{\pm0.02}$ | $77.27_{\pm11.02}$ | $100.00_{\pm0.00}$ |
| | Tail-MLP | $99.82_{\pm0.12}$ | $98.58_{\pm2.46}$ | $99.98_{\pm0.02}$ | $81.20_{\pm9.72}$ | $100.00_{\pm0.00}$ |
| | Avg-Linear | $99.57_{\pm0.24}$ | $100.00_{\pm0.00}$ | $100.00_{\pm0.00}$ | $83.00_{\pm2.03}$ | $100.00_{\pm0.00}$ |
| | NormStat:KL | $85.14_{\pm0.35}$ | $10.37_{\pm1.06}$ | $99.85_{\pm0.06}$ | $92.93_{\pm0.12}$ | $99.51_{\pm0.49}$ |
| | VecStat:KL | $99.95_{\pm0.04}$ | $99.59_{\pm0.35}$ | $99.99_{\pm0.02}$ | $92.20_{\pm0.00}$ | $100.00_{\pm0.00}$ |
| | VecStat:Cos | $100.00_{\pm0.00}$ | $95.73_{\pm0.61}$ | $99.98_{\pm0.03}$ | $94.20_{\pm0.00}$ | $100.00_{\pm0.00}$ |
| | LLM Call (0-shot) | $99.67_{\pm0.04}$ | $100.00_{\pm0.00}$ | $99.42_{\pm0.07}$ | $99.60_{\pm0.00}$ | $99.67_{\pm0.28}$ |
| | LLM Call (3-shot) | $99.14_{\pm0.18}$ | $97.15_{\pm0.35}$ | $99.43_{\pm0.07}$ | $99.87_{\pm0.12}$ | $99.51_{\pm0.49}$ |
| Qwen3-32B | Avg-MLP | $95.88_{\pm3.31}$ | $100.00_{\pm0.00}$ | $99.99_{\pm0.01}$ | $86.87_{\pm5.22}$ | $100.00_{\pm0.00}$ |
| | Tail-MLP | $99.39_{\pm0.00}$ | $100.00_{\pm0.00}$ | $99.98_{\pm0.00}$ | $96.80_{\pm0.40}$ | $99.02_{\pm0.49}$ |
| | Avg-Linear | $98.61_{\pm1.59}$ | $100.00_{\pm0.00}$ | $99.99_{\pm0.02}$ | $87.53_{\pm6.94}$ | $100.00_{\pm0.00}$ |
| | NormStat:KL | $97.93_{\pm0.04}$ | $24.59_{\pm0.35}$ | $99.78_{\pm0.06}$ | $97.93_{\pm0.12}$ | $100.00_{\pm0.00}$ |
| | VecStat:KL | $100.00_{\pm0.00}$ | $99.39_{\pm0.00}$ | $99.98_{\pm0.03}$ | $96.60_{\pm0.00}$ | $100.00_{\pm0.00}$ |
| | VecStat:Cos | $100.00_{\pm0.00}$ | $98.17_{\pm0.00}$ | $99.98_{\pm0.03}$ | $96.80_{\pm0.35}$ | $100.00_{\pm0.00}$ |
| | LLM Call (0-shot) | $86.91_{\pm0.29}$ | $100.00_{\pm0.00}$ | $98.69_{\pm0.06}$ | $96.27_{\pm0.42}$ | $100.00_{\pm0.00}$ |
| | LLM Call (3-shot) | $97.80_{\pm0.20}$ | $100.00_{\pm0.00}$ | $98.87_{\pm0.05}$ | $99.80_{\pm0.20}$ | $100.00_{\pm0.00}$ |

Table 9: Level-2 programming language classification results for all seven LLMs. Values represent per-language accuracy across nine programming languages from the Magicoder dataset.

| Model | Method | cpp | csharp | java | php | python | rust | shell | swift | typescript |
|---|---|---|---|---|---|---|---|---|---|---|
| Qwen3-1.7B | Avg-MLP | $99.97_{\pm0.03}$ | $100.00_{\pm0.00}$ | $100.00_{\pm0.00}$ | $99.96_{\pm0.06}$ | $99.89_{\pm0.05}$ | $100.00_{\pm0.00}$ | $99.92_{\pm0.14}$ | $100.00_{\pm0.00}$ | $99.94_{\pm0.07}$ |
| | Tail-MLP | $99.54_{\pm0.09}$ | $99.57_{\pm0.06}$ | $99.41_{\pm0.26}$ | $99.89_{\pm0.11}$ | $99.35_{\pm0.27}$ | $99.32_{\pm0.17}$ | $99.92_{\pm0.14}$ | $99.81_{\pm0.07}$ | $99.52_{\pm0.03}$ |
| | Avg-Linear | $99.97_{\pm0.03}$ | $99.98_{\pm0.04}$ | $100.00_{\pm0.00}$ | $99.96_{\pm0.06}$ | $99.87_{\pm0.01}$ | $100.00_{\pm0.00}$ | $99.92_{\pm0.14}$ | $100.00_{\pm0.00}$ | $99.93_{\pm0.05}$ |
| | NormStat:KL | $53.18_{\pm0.60}$ | $53.17_{\pm1.96}$ | $40.39_{\pm0.63}$ | $57.35_{\pm1.97}$ | $60.17_{\pm0.89}$ | $58.07_{\pm0.55}$ | $88.73_{\pm1.67}$ | $59.47_{\pm0.88}$ | $49.59_{\pm0.34}$ |
| | VecStat:KL | $98.83_{\pm0.18}$ | $98.98_{\pm0.39}$ | $98.21_{\pm0.15}$ | $99.81_{\pm0.23}$ | $99.06_{\pm0.09}$ | $99.68_{\pm0.14}$ | $99.52_{\pm0.00}$ | $99.81_{\pm0.12}$ | $99.26_{\pm0.20}$ |
| | VecStat:Cos | $98.51_{\pm0.19}$ | $98.78_{\pm0.37}$ | $97.79_{\pm0.19}$ | $99.85_{\pm0.17}$ | $99.06_{\pm0.14}$ | $99.35_{\pm0.15}$ | $99.84_{\pm0.14}$ | $99.55_{\pm0.11}$ | $99.02_{\pm0.29}$ |
| Qwen3-1.7B-Base | Avg-MLP | $99.97_{\pm0.03}$ | $99.96_{\pm0.04}$ | $99.97_{\pm0.03}$ | $100.00_{\pm0.00}$ | $99.89_{\pm0.05}$ | $100.00_{\pm0.00}$ | $99.92_{\pm0.14}$ | $100.00_{\pm0.00}$ | $99.96_{\pm0.04}$ |
| | Tail-MLP | $99.28_{\pm0.19}$ | $99.37_{\pm0.19}$ | $99.23_{\pm0.12}$ | $99.96_{\pm0.06}$ | $99.25_{\pm0.14}$ | $99.27_{\pm0.19}$ | $99.84_{\pm0.27}$ | $99.75_{\pm0.14}$ | $99.50_{\pm0.03}$ |
| | Avg-Linear | $99.97_{\pm0.03}$ | $99.96_{\pm0.04}$ | $99.98_{\pm0.03}$ | $100.00_{\pm0.00}$ | $99.89_{\pm0.03}$ | $100.00_{\pm0.00}$ | $99.92_{\pm0.14}$ | $100.00_{\pm0.00}$ | $99.96_{\pm0.04}$ |
| | NormStat:KL | $47.76_{\pm0.18}$ | $43.75_{\pm1.18}$ | $36.92_{\pm0.96}$ | $60.44_{\pm1.13}$ | $56.03_{\pm0.82}$ | $54.88_{\pm0.78}$ | $88.33_{\pm2.42}$ | $57.85_{\pm1.25}$ | $51.20_{\pm0.25}$ |
| | VecStat:KL | $98.29_{\pm0.13}$ | $98.37_{\pm0.53}$ | $97.00_{\pm0.21}$ | $99.66_{\pm0.11}$ | $98.63_{\pm0.09}$ | $99.45_{\pm0.12}$ | $99.68_{\pm0.14}$ | $99.64_{\pm0.12}$ | $99.17_{\pm0.19}$ |
| | VecStat:Cos | $98.27_{\pm0.17}$ | $98.23_{\pm0.50}$ | $96.91_{\pm0.05}$ | $99.63_{\pm0.17}$ | $98.84_{\pm0.09}$ | $99.63_{\pm0.17}$ | $99.68_{\pm0.14}$ | $99.58_{\pm0.07}$ | $98.54_{\pm0.30}$ |
| Llama-3.2-1B | Avg-MLP | $99.97_{\pm0.03}$ | $99.98_{\pm0.04}$ | $99.97_{\pm0.05}$ | $99.96_{\pm0.06}$ | $99.91_{\pm0.02}$ | $100.00_{\pm0.00}$ | $100.00_{\pm0.00}$ | $100.00_{\pm0.00}$ | $99.94_{\pm0.03}$ |
| | Tail-MLP | $99.21_{\pm0.24}$ | $99.08_{\pm0.16}$ | $99.41_{\pm0.10}$ | $99.85_{\pm0.17}$ | $99.25_{\pm0.26}$ | $99.21_{\pm0.32}$ | $99.84_{\pm0.27}$ | $99.75_{\pm0.09}$ | $99.58_{\pm0.11}$ |
| | Avg-Linear | $99.97_{\pm0.03}$ | $99.96_{\pm0.04}$ | $99.94_{\pm0.05}$ | $99.96_{\pm0.06}$ | $99.89_{\pm0.01}$ | $99.98_{\pm0.03}$ | $100.00_{\pm0.00}$ | $100.00_{\pm0.00}$ | $99.93_{\pm0.03}$ |
| | NormStat:KL | $44.23_{\pm1.37}$ | $39.54_{\pm2.26}$ | $35.09_{\pm1.07}$ | $49.25_{\pm1.60}$ | $38.65_{\pm0.71}$ | $54.91_{\pm0.78}$ | $90.95_{\pm1.56}$ | $55.63_{\pm1.24}$ | $34.67_{\pm1.81}$ |
| | VecStat:KL | $98.77_{\pm0.19}$ | $98.37_{\pm0.43}$ | $97.14_{\pm0.07}$ | $99.63_{\pm0.17}$ | $98.66_{\pm0.21}$ | $99.79_{\pm0.03}$ | $99.84_{\pm0.27}$ | $99.73_{\pm0.18}$ | $99.02_{\pm0.25}$ |
| | VecStat:Cos | $98.53_{\pm0.12}$ | $97.78_{\pm0.25}$ | $96.34_{\pm0.09}$ | $99.52_{\pm0.28}$ | $98.86_{\pm0.18}$ | $99.48_{\pm0.15}$ | $99.60_{\pm0.14}$ | $99.49_{\pm0.25}$ | $98.81_{\pm0.36}$ |
| Llama-3.2-1B-Instruct | Avg-MLP | $99.95_{\pm0.05}$ | $99.98_{\pm0.04}$ | $99.97_{\pm0.03}$ | $99.96_{\pm0.06}$ | $99.91_{\pm0.01}$ | $100.00_{\pm0.00}$ | $99.92_{\pm0.14}$ | $100.00_{\pm0.00}$ | $99.99_{\pm0.03}$ |
| | Tail-MLP | $99.07_{\pm0.39}$ | $99.23_{\pm0.15}$ | $99.12_{\pm0.40}$ | $99.55_{\pm0.22}$ | $99.27_{\pm0.11}$ | $99.00_{\pm0.49}$ | $99.84_{\pm0.27}$ | $99.70_{\pm0.16}$ | $99.64_{\pm0.04}$ |
| | Avg-Linear | $99.97_{\pm0.03}$ | $99.96_{\pm0.04}$ | $99.95_{\pm0.05}$ | $99.96_{\pm0.06}$ | $99.88_{\pm0.03}$ | $100.00_{\pm0.00}$ | $99.92_{\pm0.14}$ | $100.00_{\pm0.00}$ | $99.99_{\pm0.03}$ |
| | NormStat:KL | $41.34_{\pm1.45}$ | $45.89_{\pm0.57}$ | $33.32_{\pm1.51}$ | $45.23_{\pm2.25}$ | $46.42_{\pm1.32}$ | $55.80_{\pm0.67}$ | $89.76_{\pm2.52}$ | $57.64_{\pm0.72}$ | $34.05_{\pm2.20}$ |
| | VecStat:KL | $98.56_{\pm0.19}$ | $97.58_{\pm0.19}$ | $97.25_{\pm0.13}$ | $99.70_{\pm0.17}$ | $98.87_{\pm0.16}$ | $99.66_{\pm0.08}$ | $99.84_{\pm0.27}$ | $99.62_{\pm0.14}$ | $99.02_{\pm0.27}$ |
| | VecStat:Cos | $98.56_{\pm0.19}$ | $97.46_{\pm0.34}$ | $96.85_{\pm0.23}$ | $99.78_{\pm0.19}$ | $99.00_{\pm0.13}$ | $99.37_{\pm0.10}$ | $99.84_{\pm0.27}$ | $99.64_{\pm0.14}$ | $98.87_{\pm0.32}$ |
| Qwen3-4B | Avg-MLP | $99.97_{\pm0.03}$ | $99.94_{\pm0.00}$ | $100.00_{\pm0.00}$ | $99.96_{\pm0.06}$ | $99.89_{\pm0.01}$ | $100.00_{\pm0.00}$ | $100.00_{\pm0.00}$ | $100.00_{\pm0.00}$ | $99.94_{\pm0.03}$ |
| | Tail-MLP | $99.37_{\pm0.11}$ | $99.53_{\pm0.13}$ | $99.46_{\pm0.03}$ | $99.81_{\pm0.13}$ | $99.48_{\pm0.02}$ | $99.35_{\pm0.27}$ | $99.84_{\pm0.27}$ | $99.73_{\pm0.14}$ | $99.66_{\pm0.07}$ |
| | Avg-Linear | $99.97_{\pm0.03}$ | $99.96_{\pm0.07}$ | $99.98_{\pm0.03}$ | $99.96_{\pm0.06}$ | $99.89_{\pm0.02}$ | $100.00_{\pm0.00}$ | $99.92_{\pm0.14}$ | $100.00_{\pm0.00}$ | $99.93_{\pm0.05}$ |
| | NormStat:KL | $54.55_{\pm1.04}$ | $40.86_{\pm1.39}$ | $35.45_{\pm0.28}$ | $56.26_{\pm0.68}$ | $69.71_{\pm1.04}$ | $68.76_{\pm0.65}$ | $93.10_{\pm1.33}$ | $72.17_{\pm1.20}$ | $48.70_{\pm0.32}$ |
| | VecStat:KL | $99.26_{\pm0.08}$ | $98.60_{\pm0.38}$ | $97.95_{\pm0.12}$ | $99.74_{\pm0.26}$ | $99.07_{\pm0.09}$ | $99.82_{\pm0.03}$ | $99.76_{\pm0.00}$ | $99.87_{\pm0.12}$ | $99.30_{\pm0.19}$ |
| | VecStat:Cos | $98.92_{\pm0.09}$ | $98.86_{\pm0.44}$ | $98.18_{\pm0.05}$ | $99.89_{\pm0.19}$ | $99.13_{\pm0.08}$ | $99.53_{\pm0.10}$ | $99.84_{\pm0.14}$ | $99.79_{\pm0.13}$ | $99.26_{\pm0.23}$ |
| Qwen3-8B | Avg-MLP | $99.95_{\pm0.05}$ | $100.00_{\pm0.00}$ | $100.00_{\pm0.00}$ | $99.96_{\pm0.06}$ | $99.91_{\pm0.01}$ | $100.00_{\pm0.00}$ | $100.00_{\pm0.00}$ | $100.00_{\pm0.00}$ | $99.94_{\pm0.03}$ |
| | Tail-MLP | $99.61_{\pm0.16}$ | $99.69_{\pm0.18}$ | $99.47_{\pm0.12}$ | $100.00_{\pm0.00}$ | $99.39_{\pm0.22}$ | $99.53_{\pm0.10}$ | $99.92_{\pm0.14}$ | $99.75_{\pm0.07}$ | $99.61_{\pm0.10}$ |
| | Avg-Linear | $99.97_{\pm0.03}$ | $100.00_{\pm0.00}$ | $100.00_{\pm0.00}$ | $99.96_{\pm0.06}$ | $99.91_{\pm0.03}$ | $100.00_{\pm0.00}$ | $100.00_{\pm0.00}$ | $100.00_{\pm0.00}$ | $99.93_{\pm0.03}$ |
| | NormStat:KL | $47.37_{\pm0.94}$ | $38.26_{\pm1.28}$ | $27.59_{\pm0.65}$ | $60.14_{\pm1.27}$ | $67.73_{\pm0.54}$ | $63.05_{\pm0.43}$ | $91.51_{\pm0.60}$ | $62.54_{\pm1.25}$ | $49.28_{\pm0.84}$ |
| | VecStat:KL | $99.09_{\pm0.11}$ | $98.58_{\pm0.42}$ | $97.78_{\pm0.09}$ | $99.78_{\pm0.22}$ | $99.01_{\pm0.04}$ | $99.81_{\pm0.00}$ | $99.76_{\pm0.00}$ | $99.91_{\pm0.09}$ | $99.38_{\pm0.20}$ |
| | VecStat:Cos | $98.90_{\pm0.24}$ | $98.96_{\pm0.38}$ | $98.49_{\pm0.03}$ | $99.85_{\pm0.13}$ | $99.13_{\pm0.12}$ | $99.55_{\pm0.07}$ | $100.00_{\pm0.00}$ | $99.81_{\pm0.12}$ | $99.35_{\pm0.19}$ |
| Qwen3-32B | Avg-MLP | $99.91_{\pm0.03}$ | $99.98_{\pm0.04}$ | $100.00_{\pm0.00}$ | $99.96_{\pm0.06}$ | $99.91_{\pm0.03}$ | $99.97_{\pm0.06}$ | $100.00_{\pm0.00}$ | $100.00_{\pm0.00}$ | $99.96_{\pm0.04}$ |
| | Tail-MLP | $98.01_{\pm0.41}$ | $97.27_{\pm0.95}$ | $97.99_{\pm0.53}$ | $98.77_{\pm0.22}$ | $98.26_{\pm0.21}$ | $99.21_{\pm0.06}$ | $98.97_{\pm0.60}$ | $98.71_{\pm0.48}$ | $98.50_{\pm0.07}$ |
| | Avg-Linear | $99.91_{\pm0.03}$ | $99.98_{\pm0.04}$ | $100.00_{\pm0.00}$ | $99.96_{\pm0.06}$ | $99.92_{\pm0.02}$ | $99.97_{\pm0.06}$ | $100.00_{\pm0.00}$ | $100.00_{\pm0.00}$ | $99.96_{\pm0.04}$ |
| | NormStat:KL | $53.57_{\pm0.94}$ | $36.94_{\pm1.38}$ | $25.47_{\pm1.43}$ | $59.47_{\pm0.57}$ | $61.66_{\pm1.40}$ | $67.63_{\pm0.54}$ | $89.29_{\pm0.71}$ | $69.35_{\pm1.19}$ | $49.84_{\pm1.08}$ |
| | VecStat:KL | $99.01_{\pm0.11}$ | $99.35_{\pm0.34}$ | $98.72_{\pm0.03}$ | $99.89_{\pm0.11}$ | $99.29_{\pm0.06}$ | $99.90_{\pm0.05}$ | $100.00_{\pm0.00}$ | $99.85_{\pm0.09}$ | $99.75_{\pm0.16}$ |
| | VecStat:Cos | $99.18_{\pm0.14}$ | $99.53_{\pm0.19}$ | $99.01_{\pm0.10}$ | $99.89_{\pm0.11}$ | $99.50_{\pm0.04}$ | $99.81_{\pm0.13}$ | $100.00_{\pm0.00}$ | $99.87_{\pm0.12}$ | $99.72_{\pm0.09}$ |

## D.2 ADDTIONAL RESULTS FOR LEVEL-2 EXPERIMENTS

For programming languages, please refer to Table 9. For math, please refer to Table 10. For natural languages, refer to Table 11.

## D.3 CASE STUDY

Fig. 8 presents the mixed-intent prompt used in Section 3.3, constructed by concatenating code content with mathematical content. We also evaluated the reverse concatenation order (mathematics followed by code), with results shown in Fig. 9. The uncertainty quantification patterns remain consistent across both prompt orderings.

## D.4 THE EFFECT OF THE NUMBER OF LAYERS CONSIDERED

See Fig. 10.

## D.5 CALIBRATION CONVERGENCE ANALYSIS

See Fig. 11.

## D.6 THE EFFECT OF THE MAXIMUM PROMPT LENGTH

See Fig. 12.

Table 10: Level-2 mathematical subfield classification results for all seven LLMs. Values represent per-subfield accuracy across seven mathematical subfields from the Competition Math dataset.

| Model | Method | Algebra | Counting & Probability | Geometry | Intermediate Algebra | Number Theory | Prealgebra | Precalculus |
|---|---|---|---|---|---|---|---|---|
| Qwen3-1.7B | Avg-MLP | $71.78_{\pm5.57}$ | $79.48_{\pm1.91}$ | $89.98_{\pm3.46}$ | $79.33_{\pm3.96}$ | $84.02_{\pm2.56}$ | $50.00_{\pm2.29}$ | $82.79_{\pm3.37}$ |
| | Tail-MLP | $65.79_{\pm3.55}$ | $76.10_{\pm4.24}$ | $84.88_{\pm4.43}$ | $71.86_{\pm6.02}$ | $82.32_{\pm3.80}$ | $50.31_{\pm2.59}$ | $73.58_{\pm4.30}$ |
| | Avg-Linear | $64.13_{\pm3.32}$ | $75.88_{\pm0.79}$ | $89.31_{\pm1.47}$ | $81.14_{\pm3.20}$ | $79.47_{\pm4.34}$ | $52.66_{\pm2.86}$ | $79.74_{\pm1.54}$ |
| | NormStat:KL | $23.67_{\pm1.27}$ | $37.83_{\pm0.65}$ | $48.15_{\pm4.09}$ | $58.75_{\pm1.08}$ | $65.46_{\pm1.25}$ | $0.97_{\pm0.28}$ | $33.88_{\pm1.02}$ |
| | VecStat:KL | $33.29_{\pm1.39}$ | $53.71_{\pm2.00}$ | $35.88_{\pm3.28}$ | $81.88_{\pm0.58}$ | $86.26_{\pm0.62}$ | $1.46_{\pm0.05}$ | $46.07_{\pm1.82}$ |
| | VecStat:Cos | $57.66_{\pm0.70}$ | $61.50_{\pm2.16}$ | $36.07_{\pm3.37}$ | $73.58_{\pm0.83}$ | $86.92_{\pm0.38}$ | $2.22_{\pm0.32}$ | $51.22_{\pm1.61}$ |
| Qwen3-1.7B-Base | Avg-MLP | $67.96_{\pm1.83}$ | $79.55_{\pm0.67}$ | $90.53_{\pm0.55}$ | $83.93_{\pm2.34}$ | $84.02_{\pm2.62}$ | $48.85_{\pm2.19}$ | $83.47_{\pm3.46}$ |
| | Tail-MLP | $70.97_{\pm4.48}$ | $80.97_{\pm2.25}$ | $90.89_{\pm1.14}$ | $74.49_{\pm2.32}$ | $81.50_{\pm6.42}$ | $51.93_{\pm7.96}$ | $78.25_{\pm2.55}$ |
| | Avg-Linear | $66.46_{\pm4.16}$ | $75.81_{\pm2.04}$ | $88.95_{\pm1.00}$ | $81.57_{\pm1.99}$ | $81.06_{\pm2.52}$ | $51.33_{\pm2.29}$ | $79.54_{\pm1.31}$ |
| | NormStat:KL | $23.35_{\pm0.70}$ | $37.23_{\pm2.12}$ | $47.36_{\pm4.20}$ | $56.65_{\pm1.30}$ | $58.84_{\pm2.38}$ | $5.67_{\pm2.17}$ | $34.82_{\pm0.82}$ |
| | VecStat:KL | $39.48_{\pm3.28}$ | $50.86_{\pm1.72}$ | $35.88_{\pm3.37}$ | $81.31_{\pm0.65}$ | $88.12_{\pm0.58}$ | $1.52_{\pm0.52}$ | $46.41_{\pm1.24}$ |
| | VecStat:Cos | $61.24_{\pm0.90}$ | $62.62_{\pm2.31}$ | $36.00_{\pm3.55}$ | $73.46_{\pm0.80}$ | $88.18_{\pm0.28}$ | $2.66_{\pm0.67}$ | $53.59_{\pm0.82}$ |
| Llama-3.2-1B | Avg-MLP | $57.99_{\pm6.53}$ | $75.13_{\pm5.42}$ | $86.22_{\pm4.37}$ | $75.94_{\pm4.00}$ | $78.43_{\pm0.09}$ | $41.67_{\pm5.38}$ | $76.02_{\pm1.42}$ |
| | Tail-MLP | $58.38_{\pm4.88}$ | $78.35_{\pm2.40}$ | $91.20_{\pm1.28}$ | $74.44_{\pm3.57}$ | $77.45_{\pm8.21}$ | $36.26_{\pm4.71}$ | $67.01_{\pm5.55}$ |
| | Avg-Linear | $53.70_{\pm6.28}$ | $70.94_{\pm5.13}$ | $81.60_{\pm9.17}$ | $74.15_{\pm2.30}$ | $80.41_{\pm7.23}$ | $44.36_{\pm5.95}$ | $76.96_{\pm7.70}$ |
| | NormStat:KL | $19.46_{\pm3.91}$ | $16.33_{\pm1.35}$ | $38.07_{\pm2.98}$ | $37.01_{\pm3.69}$ | $80.24_{\pm2.55}$ | $0.05_{\pm0.05}$ | $27.71_{\pm1.73}$ |
| | VecStat:KL | $34.57_{\pm0.92}$ | $50.34_{\pm1.92}$ | $38.19_{\pm4.30}$ | $76.13_{\pm0.97}$ | $84.89_{\pm0.59}$ | $1.38_{\pm0.12}$ | $48.17_{\pm2.66}$ |
| | VecStat:Cos | $44.71_{\pm0.78}$ | $59.48_{\pm2.26}$ | $44.63_{\pm5.73}$ | $72.65_{\pm0.55}$ | $83.96_{\pm1.19}$ | $2.30_{\pm0.32}$ | $52.57_{\pm0.92}$ |
| Llama-3.2-1B-Instruct | Avg-MLP | $64.88_{\pm4.97}$ | $78.95_{\pm1.01}$ | $86.76_{\pm4.30}$ | $76.82_{\pm5.70}$ | $85.39_{\pm0.43}$ | $45.27_{\pm2.58}$ | $84.35_{\pm2.00}$ |
| | Tail-MLP | $73.63_{\pm4.07}$ | $79.48_{\pm2.60}$ | $88.71_{\pm2.03}$ | $78.02_{\pm2.53}$ | $83.58_{\pm0.16}$ | $51.28_{\pm3.11}$ | $83.47_{\pm1.84}$ |
| | Avg-Linear | $57.23_{\pm5.70}$ | $77.83_{\pm1.72}$ | $83.36_{\pm2.16}$ | $82.14_{\pm0.69}$ | $80.57_{\pm1.53}$ | $48.75_{\pm2.43}$ | $82.05_{\pm0.96}$ |
| | NormStat:KL | $4.38_{\pm3.24}$ | $11.69_{\pm1.57}$ | $36.13_{\pm3.19}$ | $37.34_{\pm3.24}$ | $84.40_{\pm1.00}$ | $0.26_{\pm0.12}$ | $27.98_{\pm4.39}$ |
| | VecStat:KL | $40.03_{\pm2.62}$ | $45.09_{\pm1.50}$ | $35.88_{\pm3.46}$ | $74.32_{\pm1.49}$ | $88.67_{\pm0.43}$ | $1.25_{\pm0.14}$ | $46.75_{\pm2.34}$ |
| | VecStat:Cos | $53.20_{\pm1.76}$ | $59.85_{\pm1.80}$ | $36.79_{\pm4.02}$ | $72.72_{\pm1.04}$ | $86.59_{\pm0.19}$ | $1.78_{\pm0.09}$ | $52.17_{\pm1.54}$ |
| Qwen3-4B | Avg-MLP | $73.52_{\pm1.54}$ | $79.63_{\pm4.68}$ | $89.80_{\pm2.86}$ | $79.16_{\pm1.48}$ | $80.84_{\pm0.78}$ | $51.23_{\pm3.88}$ | $84.62_{\pm2.94}$ |
| | Tail-MLP | $68.98_{\pm9.23}$ | $82.47_{\pm1.96}$ | $82.70_{\pm2.37}$ | $67.84_{\pm7.26}$ | $81.55_{\pm3.51}$ | $46.89_{\pm4.30}$ | $79.95_{\pm1.50}$ |
| | Avg-Linear | $70.64_{\pm2.89}$ | $80.07_{\pm0.34}$ | $90.59_{\pm3.57}$ | $80.02_{\pm2.61}$ | $79.97_{\pm3.76}$ | $49.53_{\pm5.57}$ | $81.91_{\pm2.12}$ |
| | NormStat:KL | $19.57_{\pm3.97}$ | $17.15_{\pm0.91}$ | $42.26_{\pm3.41}$ | $47.00_{\pm2.72}$ | $76.79_{\pm2.63}$ | $0.60_{\pm0.25}$ | $34.28_{\pm1.70}$ |
| | VecStat:KL | $29.80_{\pm3.56}$ | $51.99_{\pm1.30}$ | $35.82_{\pm3.37}$ | $82.31_{\pm1.15}$ | $89.00_{\pm0.28}$ | $1.36_{\pm0.05}$ | $54.81_{\pm3.00}$ |
| | VecStat:Cos | $62.24_{\pm2.09}$ | $61.72_{\pm2.03}$ | $36.13_{\pm3.19}$ | $73.89_{\pm1.24}$ | $88.83_{\pm0.33}$ | $1.96_{\pm0.08}$ | $53.59_{\pm1.32}$ |
| Qwen3-8B | Avg-MLP | $73.94_{\pm1.63}$ | $82.70_{\pm0.98}$ | $87.13_{\pm0.56}$ | $81.00_{\pm3.61}$ | $86.59_{\pm4.35}$ | $52.95_{\pm5.24}$ | $89.36_{\pm0.42}$ |
| | Tail-MLP | $68.43_{\pm1.81}$ | $80.52_{\pm2.03}$ | $89.92_{\pm1.91}$ | $73.87_{\pm6.12}$ | $82.92_{\pm5.55}$ | $53.58_{\pm0.64}$ | $81.10_{\pm5.72}$ |
| | Avg-Linear | $73.28_{\pm4.60}$ | $80.82_{\pm3.83}$ | $90.29_{\pm2.47}$ | $79.92_{\pm3.87}$ | $82.65_{\pm0.90}$ | $52.19_{\pm3.70}$ | $84.76_{\pm0.54}$ |
| | NormStat:KL | $21.01_{\pm6.35}$ | $19.03_{\pm1.50}$ | $40.19_{\pm3.56}$ | $46.38_{\pm0.91}$ | $73.45_{\pm4.22}$ | $0.73_{\pm0.59}$ | $31.98_{\pm1.31}$ |
| | VecStat:KL | $36.23_{\pm3.18}$ | $51.24_{\pm1.17}$ | $35.88_{\pm3.28}$ | $81.00_{\pm1.60}$ | $88.89_{\pm0.47}$ | $1.52_{\pm0.12}$ | $55.15_{\pm3.27}$ |
| | VecStat:Cos | $64.46_{\pm1.45}$ | $61.95_{\pm1.87}$ | $36.25_{\pm3.19}$ | $74.13_{\pm0.95}$ | $89.33_{\pm0.33}$ | $2.06_{\pm0.40}$ | $55.08_{\pm1.08}$ |
| Qwen3-32B | Avg-MLP | $73.25_{\pm4.75}$ | $77.83_{\pm2.89}$ | $89.01_{\pm3.74}$ | $84.14_{\pm3.27}$ | $81.88_{\pm2.64}$ | $57.92_{\pm4.22}$ | $86.65_{\pm3.02}$ |
| | Tail-MLP | $66.70_{\pm8.95}$ | $78.43_{\pm2.02}$ | $88.52_{\pm4.65}$ | $79.61_{\pm3.33}$ | $74.06_{\pm11.10}$ | $50.57_{\pm10.68}$ | $80.62_{\pm3.30}$ |
| | Avg-Linear | $71.97_{\pm5.76}$ | $79.93_{\pm3.82}$ | $87.86_{\pm0.92}$ | $82.33_{\pm3.88}$ | $85.06_{\pm0.72}$ | $52.82_{\pm4.70}$ | $87.53_{\pm1.89}$ |
| | NormStat:KL | $31.99_{\pm2.02}$ | $21.72_{\pm0.85}$ | $40.56_{\pm3.48}$ | $44.61_{\pm1.14}$ | $75.53_{\pm2.56}$ | $0.13_{\pm0.05}$ | $31.44_{\pm1.47}$ |
| | VecStat:KL | $55.76_{\pm1.11}$ | $58.50_{\pm2.04}$ | $35.94_{\pm3.28}$ | $77.80_{\pm1.04}$ | $89.87_{\pm0.34}$ | $1.46_{\pm0.12}$ | $52.78_{\pm0.59}$ |
| | VecStat:Cos | $71.16_{\pm0.46}$ | $64.42_{\pm2.16}$ | $37.04_{\pm3.20}$ | $73.32_{\pm0.82}$ | $89.49_{\pm0.87}$ | $3.00_{\pm0.72}$ | $58.47_{\pm1.31}$ |

## D.7 ROBUSTNESS TO ADVERSARIAL ATTACK

We evaluate the robustness of different classification methods against adversarial perturbations on the MATH500 dataset. We construct adversarial examples by prompting GPT-5-mini to inject misleading superficial features while preserving the mathematical content (see Table 12 for the generation template and Table 13 for example outputs).

Table 14 shows classification accuracy on original and adversarially perturbed prompts. All methods are trained or calibrated on clean data only. Overall, all methods degrade under adversarial rephrasing, demonstrating the challenge of this attack. However, the results reveal interesting robustness patterns.

Training-based methods show divergent behaviors. Tail-MLP maintains robustness (retaining most of its accuracy), while Avg-MLP fails catastrophically with near-zero accuracy across all models. This stark difference indicates that final-token representations in decoder-only LLMs—which attend to all previous tokens—are more resistant to scattered misleading signals than averaged embeddings, which are easily corrupted by adversarial tokens distributed throughout the prompt.

Training-free statistical methods demonstrate different levels of robustness. NormStat shows notable resilience, retaining 58-96% of its original accuracy under attack. For instance, on Llama-3.2-1B-Instruct, it drops only from 96.2% to 92.4%. In contrast, VecStat variants are more vulnerable (e.g., VecStat:KL drops from 87.9% to 44.2% on Llama-3.2-1B-Instruct). This aligns with our method design: NormStat's reliance on radial statistics makes it relatively invariant to directional perturbations induced by stylistic rephrasing, while VecStat's directional information makes it more sensitive to subtle changes in representation geometry.

The robustness of NormStat (using radial statistics) and Tail-MLP (using final-token features) suggests that different defense mechanisms—statistical aggregation versus extensive context aggrega-

Table 11: Level-2 natural language classification results for all seven LLMs. Values represent per-language accuracy across five natural languages from the Aya dataset.

| Model | Method | English | Japanese | Moroccan Arabic | Sinhala | Tamil |
|---|---|---|---|---|---|---|
| Qwen3-1.7B | Avg-MLP | $99.74_{\pm0.12}$ | $99.99_{\pm0.02}$ | $99.97_{\pm0.03}$ | $99.99_{\pm0.02}$ | $99.94_{\pm0.04}$ |
| | Tail-MLP | $99.49_{\pm0.15}$ | $100.00_{\pm0.00}$ | $99.97_{\pm0.03}$ | $99.93_{\pm0.06}$ | $99.94_{\pm0.10}$ |
| | Avg-Linear | $99.79_{\pm0.11}$ | $100.00_{\pm0.00}$ | $99.97_{\pm0.03}$ | $99.99_{\pm0.02}$ | $99.94_{\pm0.04}$ |
| | NormStat:KL | $82.26_{\pm1.39}$ | $81.42_{\pm0.81}$ | $85.78_{\pm0.86}$ | $99.87_{\pm0.12}$ | $99.52_{\pm0.12}$ |
| | VecStat:KL | $96.27_{\pm0.34}$ | $99.83_{\pm0.03}$ | $99.97_{\pm0.00}$ | $99.99_{\pm0.02}$ | $99.93_{\pm0.06}$ |
| | VecStat:Cos | $98.09_{\pm0.10}$ | $99.82_{\pm0.05}$ | $99.99_{\pm0.02}$ | $99.99_{\pm0.02}$ | $99.94_{\pm0.04}$ |
| Qwen3-1.7B-Base | Avg-MLP | $99.64_{\pm0.15}$ | $100.00_{\pm0.00}$ | $99.94_{\pm0.04}$ | $99.99_{\pm0.02}$ | $99.96_{\pm0.02}$ |
| | Tail-MLP | $99.67_{\pm0.12}$ | $100.00_{\pm0.00}$ | $99.97_{\pm0.03}$ | $99.98_{\pm0.02}$ | $99.87_{\pm0.03}$ |
| | Avg-Linear | $99.66_{\pm0.16}$ | $99.98_{\pm0.04}$ | $99.94_{\pm0.04}$ | $99.99_{\pm0.02}$ | $99.92_{\pm0.08}$ |
| | NormStat:KL | $82.58_{\pm1.88}$ | $83.08_{\pm1.35}$ | $73.59_{\pm0.57}$ | $99.83_{\pm0.09}$ | $98.43_{\pm0.17}$ |
| | VecStat:KL | $97.41_{\pm0.13}$ | $99.84_{\pm0.05}$ | $99.97_{\pm0.03}$ | $99.99_{\pm0.02}$ | $99.93_{\pm0.06}$ |
| | VecStat:Cos | $98.46_{\pm0.13}$ | $99.83_{\pm0.09}$ | $100.00_{\pm0.00}$ | $99.99_{\pm0.02}$ | $99.94_{\pm0.04}$ |
| Llama-3.2-1B | Avg-MLP | $99.63_{\pm0.21}$ | $100.00_{\pm0.00}$ | $99.97_{\pm0.03}$ | $99.99_{\pm0.02}$ | $99.96_{\pm0.02}$ |
| | Tail-MLP | $99.60_{\pm0.18}$ | $100.00_{\pm0.00}$ | $99.97_{\pm0.03}$ | $99.87_{\pm0.00}$ | $99.71_{\pm0.28}$ |
| | Avg-Linear | $99.61_{\pm0.08}$ | $99.99_{\pm0.02}$ | $99.97_{\pm0.03}$ | $99.99_{\pm0.02}$ | $99.96_{\pm0.02}$ |
| | NormStat:KL | $77.26_{\pm6.21}$ | $57.36_{\pm3.14}$ | $98.56_{\pm0.11}$ | $99.67_{\pm0.20}$ | $99.82_{\pm0.19}$ |
| | VecStat:KL | $96.12_{\pm0.39}$ | $99.92_{\pm0.02}$ | $99.96_{\pm0.02}$ | $100.00_{\pm0.00}$ | $99.96_{\pm0.02}$ |
| | VecStat:Cos | $98.76_{\pm0.07}$ | $99.98_{\pm0.02}$ | $99.96_{\pm0.02}$ | $99.99_{\pm0.02}$ | $99.94_{\pm0.04}$ |
| Llama-3.2-1B-Instruct | Avg-MLP | $99.72_{\pm0.10}$ | $99.99_{\pm0.02}$ | $99.94_{\pm0.04}$ | $99.99_{\pm0.02}$ | $99.97_{\pm0.03}$ |
| | Tail-MLP | $99.58_{\pm0.15}$ | $100.00_{\pm0.00}$ | $99.96_{\pm0.02}$ | $99.94_{\pm0.07}$ | $99.82_{\pm0.20}$ |
| | Avg-Linear | $99.67_{\pm0.12}$ | $100.00_{\pm0.00}$ | $99.94_{\pm0.04}$ | $99.99_{\pm0.02}$ | $99.96_{\pm0.02}$ |
| | NormStat:KL | $79.47_{\pm5.07}$ | $42.79_{\pm2.16}$ | $98.54_{\pm0.18}$ | $99.86_{\pm0.13}$ | $99.86_{\pm0.07}$ |
| | VecStat:KL | $96.04_{\pm0.39}$ | $99.84_{\pm0.13}$ | $99.98_{\pm0.02}$ | $99.99_{\pm0.02}$ | $99.96_{\pm0.02}$ |
| | VecStat:Cos | $98.59_{\pm0.12}$ | $99.98_{\pm0.02}$ | $99.94_{\pm0.04}$ | $99.99_{\pm0.02}$ | $99.94_{\pm0.04}$ |
| Qwen3-4B | Avg-MLP | $99.71_{\pm0.10}$ | $100.00_{\pm0.00}$ | $99.94_{\pm0.04}$ | $99.99_{\pm0.02}$ | $99.96_{\pm0.02}$ |
| | Tail-MLP | $99.60_{\pm0.15}$ | $100.00_{\pm0.00}$ | $99.97_{\pm0.03}$ | $99.99_{\pm0.02}$ | $99.99_{\pm0.02}$ |
| | Avg-Linear | $99.74_{\pm0.12}$ | $100.00_{\pm0.00}$ | $99.94_{\pm0.04}$ | $99.99_{\pm0.02}$ | $99.96_{\pm0.02}$ |
| | NormStat:KL | $84.60_{\pm0.34}$ | $90.29_{\pm1.27}$ | $90.53_{\pm0.61}$ | $99.84_{\pm0.07}$ | $99.11_{\pm0.54}$ |
| | VecStat:KL | $93.99_{\pm0.82}$ | $99.77_{\pm0.09}$ | $100.00_{\pm0.00}$ | $99.99_{\pm0.02}$ | $99.93_{\pm0.06}$ |
| | VecStat:Cos | $97.73_{\pm0.12}$ | $99.88_{\pm0.10}$ | $100.00_{\pm0.00}$ | $99.99_{\pm0.02}$ | $99.94_{\pm0.04}$ |
| Qwen3-8B | Avg-MLP | $99.70_{\pm0.06}$ | $100.00_{\pm0.00}$ | $99.94_{\pm0.04}$ | $99.99_{\pm0.02}$ | $99.97_{\pm0.00}$ |
| | Tail-MLP | $99.53_{\pm0.15}$ | $100.00_{\pm0.00}$ | $99.97_{\pm0.03}$ | $99.99_{\pm0.02}$ | $99.92_{\pm0.08}$ |
| | Avg-Linear | $99.81_{\pm0.05}$ | $100.00_{\pm0.00}$ | $99.94_{\pm0.04}$ | $99.99_{\pm0.02}$ | $99.96_{\pm0.02}$ |
| | NormStat:KL | $85.06_{\pm0.85}$ | $92.03_{\pm1.32}$ | $76.76_{\pm0.77}$ | $99.92_{\pm0.04}$ | $96.69_{\pm2.25}$ |
| | VecStat:KL | $96.27_{\pm0.38}$ | $99.82_{\pm0.05}$ | $100.00_{\pm0.00}$ | $100.00_{\pm0.00}$ | $99.93_{\pm0.06}$ |
| | VecStat:Cos | $98.66_{\pm0.13}$ | $99.83_{\pm0.09}$ | $100.00_{\pm0.00}$ | $99.99_{\pm0.02}$ | $99.94_{\pm0.04}$ |
| Qwen3-32B | Avg-MLP | $99.76_{\pm0.17}$ | $99.99_{\pm0.02}$ | $99.94_{\pm0.04}$ | $99.99_{\pm0.02}$ | $99.96_{\pm0.04}$ |
| | Tail-MLP | $99.31_{\pm0.02}$ | $100.00_{\pm0.00}$ | $99.96_{\pm0.02}$ | $99.99_{\pm0.02}$ | $99.94_{\pm0.04}$ |
| | Avg-Linear | $99.87_{\pm0.06}$ | $100.00_{\pm0.00}$ | $99.94_{\pm0.04}$ | $99.98_{\pm0.02}$ | $99.93_{\pm0.03}$ |
| | NormStat:KL | $82.00_{\pm0.74}$ | $72.11_{\pm1.63}$ | $96.17_{\pm0.40}$ | $99.48_{\pm0.10}$ | $98.32_{\pm0.35}$ |
| | VecStat:KL | $94.08_{\pm0.27}$ | $99.68_{\pm0.18}$ | $100.00_{\pm0.00}$ | $99.99_{\pm0.02}$ | $99.94_{\pm0.04}$ |
| | VecStat:Cos | $98.20_{\pm0.12}$ | $99.78_{\pm0.07}$ | $100.00_{\pm0.00}$ | $99.99_{\pm0.02}$ | $99.96_{\pm0.02}$ |

tion—can both provide resilience against adversarial attacks in intent classification systems. Furthermore, the superior uncertainty calibration of training-free methods (demonstrated in Section 3.3) could potentially serve as an adversarial detector. While our training-free methods (especially NormStat:KL) show competitive robustness, we acknowledge that achieving full adversarial robustness remains an open challenge that warrants further exploration.

# E    ADDITIONAL DISCUSSION

## E.1    MORE RELATED WORKS

**LLM Routing**    Early work on LLM routing either ensembles outputs from multiple models (Jiang et al., 2023; Wang et al., 2023a) or uses cascades that query models sequentially by capability (Aggarwal et al., 2023; Chen et al., 2023; Yue et al., 2023), but both incur high latency and cost due to multiple calls per query. Subsequent approaches train learned routers–model-based predictors–that estimate per-query quality or cost and select the target LLM (Hari & Thomson, 2023; Stripelis et al.,

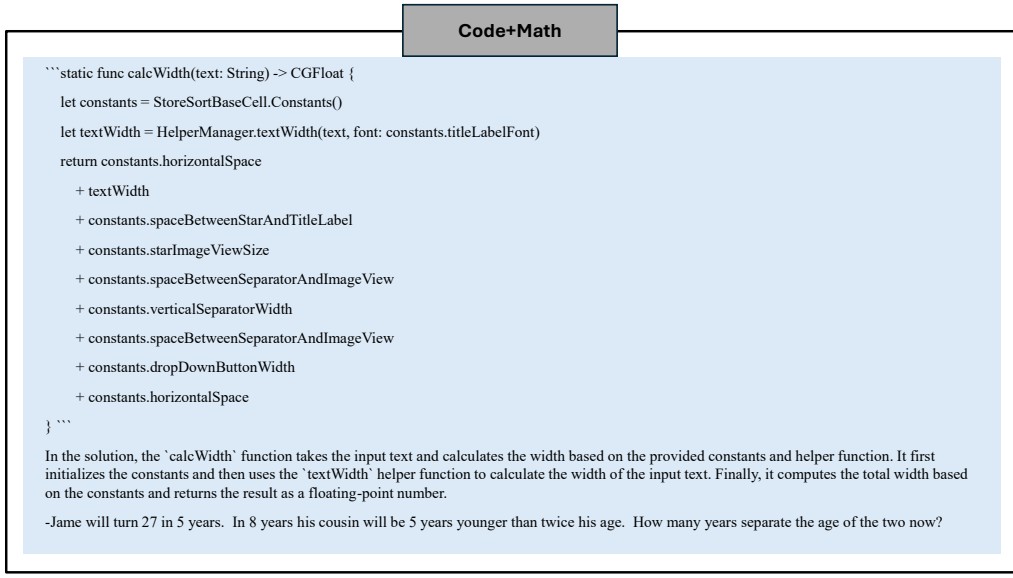

Figure 8: Mixed-intent prompt example combining Swift code (width calculation function) and a mathematical problem used for uncertainty quantification analysis.

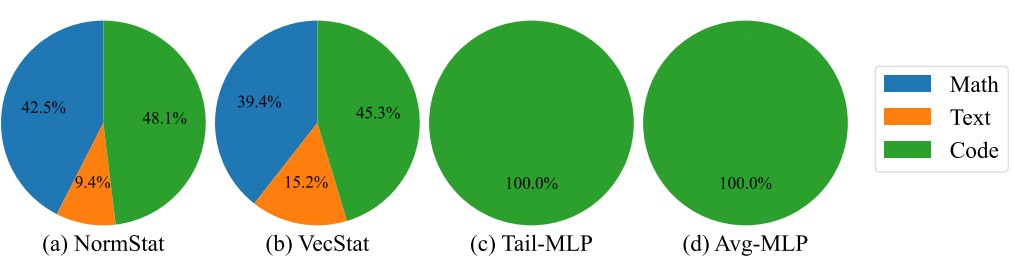

(a) NormStat    (b) VecStat    (c) Tail-MLP    (d) Avg-MLP

Figure 9: Uncertainty quantification results on mixed-intent prompt with reversed concatenation order (mathematical contents followed by code contents) on Qwen3-1.7B-Base.

2024; Feng et al., 2024; Dekoninck et al., 2024; Somerstep et al., 2025; Jitkrittum et al., 2025); these reduce unnecessary calls but introduce nontrivial training and maintenance overhead. Meanwhile, there are also training-free routers which choose among LLMs using lightweight ranking or budget-aware criteria (Zhao et al., 2024; Wu & Silwal, 2025). In contrast, our training-free statistical method is cheaper still because it operates entirely within a single LLM's prefill: we compute simple statistics of internal activations to obtain fast, calibrated intent probabilities that serve as an efficient router without extra forward passes or router training.

### E.2 WHEN ARE TRAINING-FREE METHODS PREFERABLE?

- High-throughput, multi-tenant systems. A provider may host a large number of routers (per product, per customer, or per domain), where the intent label space evolves over time as new tools, experts, or domains are introduced. In such settings, every change in the class set would require retraining an MLP head, whereas VecStat/NormStat only require adding or removing per-class statistics—keeping the adaptation cost essentially constant as the system scales.

- Untrusted or safety-critical deployments. In systems that must handle untrusted inputs (e.g. public-facing assistants), reliable uncertainty quantification is crucial for detecting malicious or

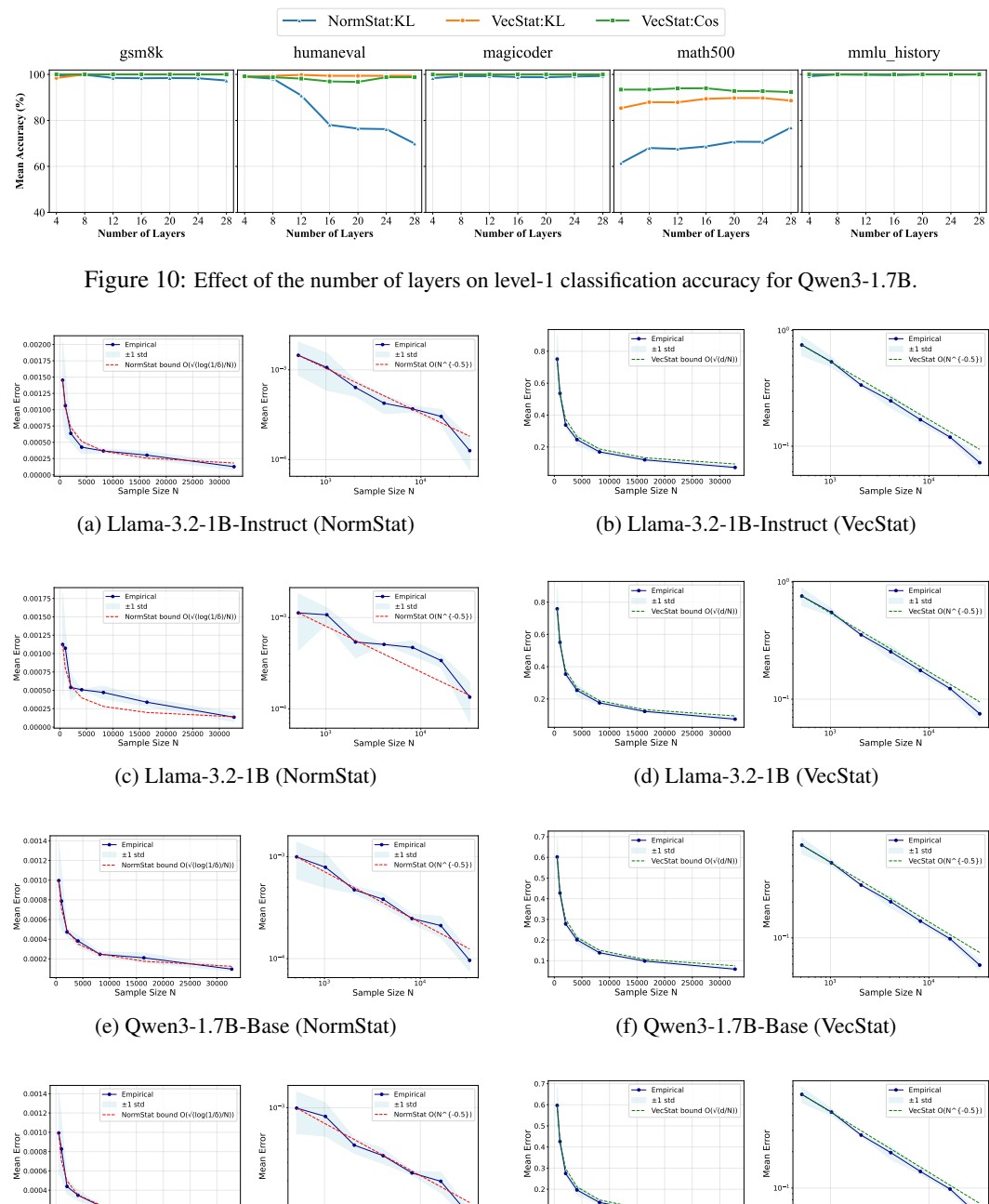

Figure 10: Effect of the number of layers on level-1 classification accuracy for Qwen3-1.7B.

(a) Llama-3.2-1B-Instruct (NormStat)   (b) Llama-3.2-1B-Instruct (VecStat)

(c) Llama-3.2-1B (NormStat)   (d) Llama-3.2-1B (VecStat)

(e) Qwen3-1.7B-Base (NormStat)   (f) Qwen3-1.7B-Base (VecStat)

(g) Qwen3-1.7B (NormStat)   (h) Qwen3-1.7B (VecStat)

Figure 11: Calibration convergence analysis for different models and methods on the MagiCoder dataset. Each subplot shows both linear and log-log scales comparing empirical results with theoretical bounds. Norm-Stat (norm method) uses dimension-free bounds while VecStat (projection method) uses dimension-dependent bounds.

out-of-distribution prompts, and human interpretability is needed for post-hoc audits (e.g., to assess potential fairness issues). Our training-free methods directly expose calibrated per-class statistics, which can be inspected and monitored without the additional modeling and engineering complexity required to obtain well-calibrated uncertainty estimates from MLP heads.

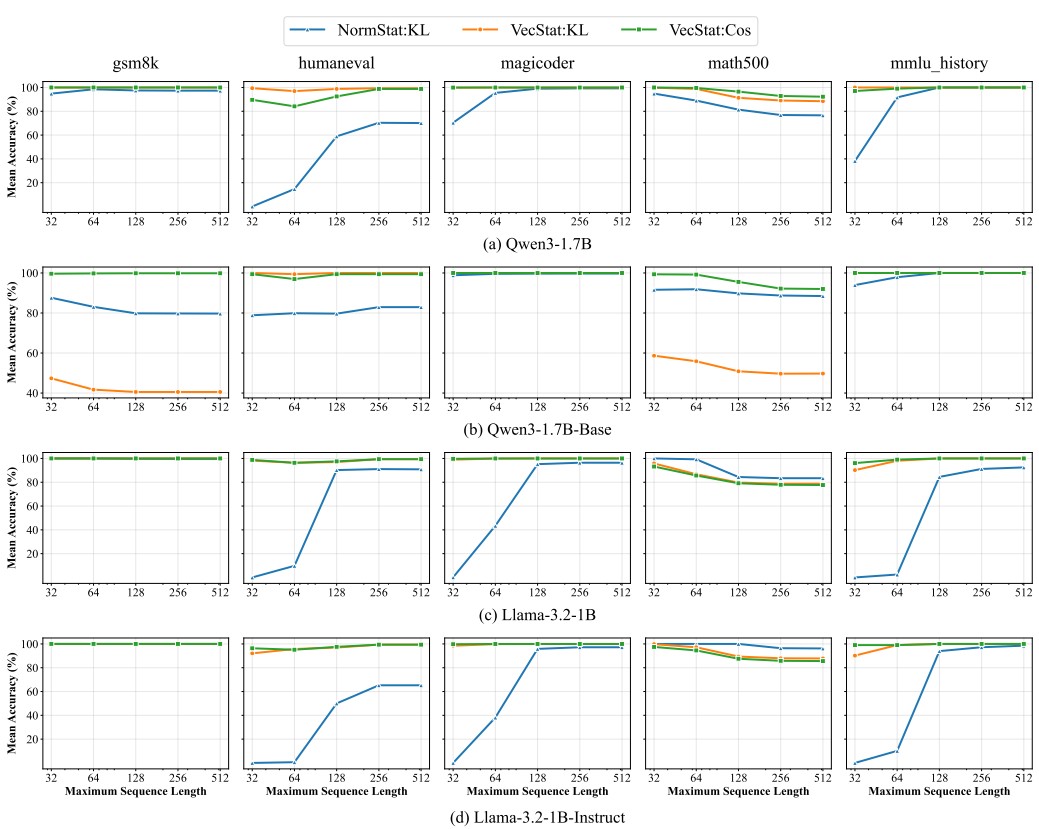

Figure 12: Effect of the maximum prompt length on level-1 classification accuracy for Qwen3-1.7B (a), Qwen3-1.7B-Base (b), Llama-3.2-1B (c), and Llama-3.2-1B-Instruct (d)..

Table 12: Adversarial prompt generation template used with GPT-5-mini. Original mathematics problems are inserted at {prompt} to generate superficially code-like versions.

---

**Prompt Template**

---

Rewrite the following math problem to make an intent classifier predict "code", while keeping it obviously a math problem to any human reader.
Original problem: {**prompt**}

Core Principle:
A human should immediately recognize this as a math problem that requires mathematical reasoning to solve. The "coding disguise" should be purely superficial - in the framing, vocabulary, and presentation style, not in the underlying task.
Requirements:

1. PRESERVE MATHEMATICAL CLARITY:

   - The numbers and their relationships should be immediately visible
   - The mathematical operation(s) needed should be obvious to a human
   - A human should think "this is clearly a math problem with coding window dressing"
   - Don't bury the math in convoluted programming scenarios

2. ADD CODING VOCABULARY (surface-level):

   - Use programming terms for framing: "function", "return", "parameter", "variable"
   - Mention a programming language: Python, Java, C++, etc.
   - Use coding-style notation where natural: = instead of "equals", * for multiply
   - Frame as function input/output rather than "find" or "calculate"

3. WHAT TO AVOID (these make it unclear to humans):

   ✗ Complex programming scenarios that obscure the math
   ✗ Adding algorithmic steps unrelated to the core math
   ✗ Nested function calls or complicated code logic
   ✗ Programming-specific complications (edge cases, type checking, etc.)
   ✗ Data structures that hide the numbers (nested arrays, linked lists, etc.)

4. THE ATTACK SHOULD BE:

   - Stylistic: Change how it's presented, not what's being asked
   - Transparent: Humans immediately see through the coding framing
   - Keyword-based: Inject terms that classifiers might key on
   - Shallow: Surface-level reframing only

5. LENGTH AND NATURALNESS:

   - Keep it concise: 1.5x to 2x original length
   - Should read like a lazy programmer asking a math question
   - Natural phrasing: "A function computes..." not "This is code question..."

6. TEST YOUR REWRITE:

   - Ask yourself: "If I saw this on a forum, would I immediately know to solve it with basic math, or would I be confused about what's being asked?"
   - If confused → too obscure, revise to be clearer
   - If obvious math → good adversarial example!

OVERALL GOAL: Create examples where:

- Humans: "That's obviously a math problem, why is it using coding terms?" ✓

- Classifier:"I see 'function', 'return', 'algorithm' → must be CODE" ✗

Your rewritten problem:

---

Table 13: Examples of adversarial prompt generated for MATH500 dataset.

| Original Prompt | Adversarial Prompt |
|---|---|
| Let $f(x) = 2x - 3$ and $g(x) = x + 1$. What is the value of $g(f(5) - 1)$? | In Python-style notation, two functions are defined as follows:
$f(x) = 2 * x - 3$ # returns 2 times x minus 3
$g(x) = x + 1$ # returns x plus 1
If the program evaluates $g(f(5) - 1)$, what integer value is returned? |
| Find the product of the $y$-coordinates of all the distinct solutions $(x, y)$ for the two equations $y = x^2 - 8$ and $y^2 = -5x + 44$. | You are writing a small Python-style function find_product() with no parameters that solves the system
y = x*x - 8
y*y = -5*x + 44
for all distinct solution pairs (x,y), and then returns the product of the y-values. What integer should find_product() return? |
| What is the result when the greatest common factor of 6432 and 132 is increased by 11? | In Python-like pseudocode:
def compute():
    gcd(6432, 132)
    return base + 11
What integer should compute() return? |
| The Greek army contained two types of soldiers: the upper class and the lower class soldiers. If there were a total of 5 upper class soldiers, and 10 lower class soldiers in a certain part of Athens, and the battle of Thermopylae demands a force of 4 upper class soldiers and 8 lower class soldiers, how many different battalions can be sent? | In Python, write a function battalion_count(upper= 5, lower= 10, need_upper= 4, need_lower= 8) that returns the number of distinct battalions formed by choosing need_upper soldiers from the upper class and need_lower soldiers from the lower class. In formula form: return C(upper, need_upper) * C(lower, need_lower) (i.e. (5 choose 4) * (10 choose 8)). With the parameters shown, what integer should battalion_count(...) return? |
| The volume of a cone is given by the formula $V = \frac{1}{3}Bh$, where $B$ is the area of the base and $h$ is the height. The area of the base of a cone is 30 square units, and its height is 6.5 units. What is the number of cubic units in its volume? | In Python, a function computes the volume of a cone using the formula V = 1/3 * B * h. If the function is defined with parameters B = 30 (area of base, in square units) and h = 6.5 (height, in units):
def cone_volume(B=30, h=6.5):
    return 1/3 * B * h
What numeric value (in cubic units) should cone_volume return? |
| If $\mathbf{a}$ and $\mathbf{b}$ are vectors such that $\|\mathbf{a}\| = 2$, $\|\mathbf{b}\| = 7$, and $$\mathbf{a} \times \mathbf{b} = \begin{pmatrix} 3 \\ 2 \\ 6 \end{pmatrix},$$ then find the smallest possible angle between $\mathbf{a}$ and $\mathbf{b}$, in degrees. | You're writing a Python function angle_between(a, b) that should return the smallest angle between two 3D vectors a and b, in degrees. For the parameters you are given:
- $\|\mathbf{a}\| = 2$
- $\|\mathbf{b}\| = 7$
- $\mathbf{a} \times \mathbf{b} = (3, 2, 6)$
Using these inputs, what numeric value should angle_between(a, b) return? (Give the smallest possible angle in degrees.) |

Table 14: Level-1 classification accuracy on original and adversarially perturbed MATH500 prompts. All methods are calibrated or trained on clean data only.

| Model | Method | math500 | adv_math500 |
|---|---|---|---|
| Llama-3.2-1B | Avg-MLP | $64.33_{\pm8.33}$ | $1.53_{\pm1.01}$ |
| | Tail-MLP | $98.93_{\pm0.42}$ | $94.20_{\pm2.16}$ |
| | NormStat:KL | $83.40_{\pm0.00}$ | $77.07_{\pm0.12}$ |
| | VecStat:KL | $78.80_{\pm0.12}$ | $12.87_{\pm0.64}$ |
| | VecStat:Cos | $77.60_{\pm0.20}$ | $9.33_{\pm0.23}$ |
| Llama-3.2-1B-Instruct | Avg-MLP | $72.40_{\pm2.91}$ | $0.73_{\pm0.12}$ |
| | Tail-MLP | $94.53_{\pm0.42}$ | $82.33_{\pm3.06}$ |
| | NormStat:KL | $96.20_{\pm0.00}$ | $92.40_{\pm0.00}$ |
| | VecStat:KL | $87.87_{\pm0.11}$ | $44.20_{\pm0.69}$ |
| | VecStat:Cos | $85.60_{\pm0.00}$ | $28.07_{\pm0.23}$ |
| Qwen3-32B | Avg-MLP | $86.87_{\pm5.22}$ | $0.00_{\pm0.00}$ |
| | Tail-MLP | $96.80_{\pm0.40}$ | $23.13_{\pm9.07}$ |
| | NormStat:KL | $97.93_{\pm0.12}$ | $57.73_{\pm0.12}$ |
| | VecStat:KL | $96.60_{\pm0.00}$ | $8.33_{\pm0.23}$ |
| | VecStat:Cos | $96.80_{\pm0.35}$ | $12.20_{\pm0.53}$ |

