# OpenReview forum: "Fast Intent Classification for LLM Routing via Statistical Analysis of Representations"
_ICLR.cc/2026/Conference — ICLR 2026 Conference Withdrawn Submission_

### Official Review · Reviewer_crfd · 2025-10-29

**Soundness:** 4
**Presentation:** 4
**Contribution:** 3
**Rating:** 8
**Confidence:** 4

**Summary:**

This paper introduces two novel training-free methods, VecStat and NormStat, for intent classification in LLMs. The main idea is to leverage statistical analysis of internal model representations during the LLM's prefill phase to categorize user prompts into predefined classes (e.g., mathematics, coding, general text). This classification enables efficient routing of prompts to specialized LLMs, aiming to improve both accuracy and computational efficiency in production systems.

The authors systematically compare their training-free approaches against traditional training-based methods, such as MLP classifiers, across various LLMs (1B to 32B parameters) and diverse datasets, encompassing both coarse-grained (Level-1) and fine-grained (Level-2) intent classification tasks. Their empirical evaluation highlights scenarios where training-free methods are most effective, particularly NormStat for coarse-grained classification and VecStat for fine-grained tasks.

A key benefit emphasized is the inherent uncertainty quantification provided by their statistical methods, which often outperforms the overconfident predictions of training-based models on ambiguous prompts. The paper also provides theoretical analyses to explain when each method is preferable, considering factors like feature directionality and magnitude, and discusses computational, memory, and calibration costs.

**Strengths:**

This approach of training-free methods addresses key challenges in LLM routing, such as the need for rapid updates when classes change and the avoidance of extensive retraining, which can be computationally expensive and time-consuming. The methods operate during the prefill phase with negligible extra cost. This is crucial for deployment in production LLM systems where inference latency is a major concern. Another notable strength is the ability of NormStat and VecStat to provide well-calibrated class probabilities and uncertainty estimates, even on mixed-intent prompts. This is a considerable advantage over many training-based methods that often require post-hoc calibration to mitigate overconfidence. The visual evidence in Figure 2 clearly demonstrates this benefit. The paper presents an extensive empirical analysis across seven LLMs (from 1B to 32B parameters) and multiple benchmark datasets for both coarse-grained and fine-grained classification. This thorough evaluation provides valuable insights into the performance characteristics of each method under different conditions.

**Weaknesses:**

While the methods are training-free for classification, their effectiveness inherently relies on the quality and discriminative power of the LLM's internal representations. If these representations are not well-separated for different intents, the statistical methods might struggle, as suggested by the lower performance of NormStat on certain fine-grained tasks (e.g., mathematical subfield classification). For mathematical subfield classification (a Level-2 task), both NormStat and VecStat fall short compared to training-based approaches (Table 2, page 7 and Table 8, page 24). This suggests limitations for tasks requiring more complex, non-linear discrimination that supervised training can capture. While the methods are "training-free" for the classifier, they still require "calibration data" to compute per-class baselines. It would be beneficial to explicitly clarify the differences in data requirements and costs (e.g., annotation effort, quantity) between "calibration data" for their methods and "training data" for traditional classifiers to fully convey the practical advantages.

**Questions:**

1. How do VecStat and NormStat adapt to entirely new, unseen intent classes? Do the authors think of a hierarchical approach where coarse-grained (Level-1) intents are identified by statistical methods, and then fine-grained (Level-2) or novel intents are handled by a fallback mechanism or a different classification strategy?
2.  The theoretical analysis for VecStat assumes a diagonal covariance matrix. How might the performance or theoretical guarantees of VecStat change if the covariance structure is more complex and non-diagonal, reflecting richer correlations between features?
3. How robust are these statistical methods to prompts designed to intentionally mislead the intent classifier? Given the training-free nature, are there specific vulnerabilities to adversarial attacks, and have the authors considered any countermeasures?

---

> ### Author Response · Authors · 2025-11-21
>
> Dear Reviewer crfd,
>
> Thank you for the detailed and insightful review, as well as the very encouraging assessment of the paper’s soundness, presentation, and contribution. We have updated the paper accordingly, and all revisions are highlighted in blue. Below, we respond to the identified weaknesses and questions, and the corresponding clarifications and new experiments have been incorporated into the revised manuscript.
>
> **W1: While the methods are training-free for classification, their effectiveness inherently relies on the quality and discriminative power of the LLM's internal representations. If these representations are not well-separated for different intents, the statistical methods might struggle, as suggested by the lower performance of NormStat on certain fine-grained tasks (e.g., mathematical subfield classification). For mathematical subfield classification (a Level-2 task), both NormStat and VecStat fall short compared to training-based approaches (Table 2, page 7 and Table 8, page 24). This suggests limitations for tasks requiring more complex, non-linear discrimination that supervised training can capture. While the methods are "training-free" for the classifier, they still require "calibration data" to compute per-class baselines. It would be beneficial to explicitly clarify the differences in data requirements and costs (e.g., annotation effort, quantity) between "calibration data" for their methods and "training data" for traditional classifiers to fully convey the practical advantages.**
>
> Thank you for the insightful comments. We agree that clarifying how calibration differs from traditional training, and discussing the practical implications for intent classification, will better highlight the strengths and limitations of our approach. Overall, the major contribution of this paper lies in providing a systematic comparison between training-free and training-based intent classification methods, and the goal is to show when one approach is better than the other.
>
> Both training-based and training-free methods require the LLM prefill stage to extract features—either for collecting statistics (for calibration) or generating training embeddings (for training). Thus, both approaches add negligible latency to intent classification, as their computational overhead is minimal compared to the LLM prefill phase itself. However, the fundamental difference lies in their adaptation mechanisms for evolving classification systems.
>
> Training-free methods have no trainable parameters; they require only calibration, where statistical features are collected from representative samples of each class. By contrast, training-based methods require supervised training to optimize its trainable parameters for intent discrimination. While initial training time remains trivial, practical challenges emerge in production systems that frequently incorporate new intent classes.
>
> When adding a new class, training-free methods maintain constant cost—simply computing statistics for the new class data. On the contrary, training-based methods face a trade-off: either (i) regenerate training embeddings for all classes (both existing and new) from scratch to enable retraining—a process whose cost grows linearly with the total number of classes, or (ii) cache all training embeddings to enable rapid retraining at the expense of storage. In Figure 3 of the revised manuscript, we quantify the storage implications of option (ii) for Qwen3-32B: as the number of intent classes gradually increases to 1,000, the total storage requirement reaches approximately 20GB (primarily for cached embeddings), although the MLP weights alone require only 6MB. By comparison, NormStat requires only 1.7MB for the same 1,000 classes—a difference of over an order of magnitude.
>
> These characteristics suggest that training-free methods are particularly well-suited for dynamic intent classification systems where new classes are frequently added and storage efficiency is valued, while training-based methods are only effective when the class definitions remain relatively stable and sufficient storage resources are available.

---

> ### Author Response · Authors · 2025-11-21
>
> **Q1. How do VecStat and NormStat adapt to entirely new, unseen intent classes? Do the authors think of a hierarchical approach where coarse-grained (Level-1) intents are identified by statistical methods, and then fine-grained (Level-2) or novel intents are handled by a fallback mechanism or a different classification strategy?**
>
> Thank you for raising this important question.
>
> **Adapting to new intent classes:** Our training-free methods (VecStat and NormStat) offer significant advantages when adapting to entirely new, unseen intent classes. To accommodate a new class, we simply need to collect statistics from representative samples of that class—no retraining is required. This adaptation process requires constant time regardless of the number of existing classes, enabling rapid and efficient adaptation to evolving classification systems. In contrast, training-based methods, which require retraining in such scenarios, face substantial challenges when incorporating new classes, as discussed in our response to W1.
>
> **Hierarchical approach:** We strongly agree that a hierarchical classification strategy is a promising direction for future work. Our empirical results already provide evidence supporting this approach: statistical methods excel at coarse-grained (Level-1) classification and provide well-calibrated uncertainty estimates (Section 3.3). This uncertainty quantification capability makes them ideal for detecting ambiguous or novel prompts that should be routed to more specialized models or fallback mechanisms. A system could leverage the computational efficiency of statistical methods for initial coarse-grained classification, then invoke more sophisticated methods only when needed for fine-grained distinctions or novel intent detection. Thank you for this valuable suggestion—we believe it represents an important avenue for future research in scalable intent classification systems.
>
>
> **Q2.The theoretical analysis for VecStat assumes a diagonal covariance matrix. How might the performance or theoretical guarantees of VecStat change if the covariance structure is more complex and non-diagonal, reflecting richer correlations between features?**
>
> Thanks for bringing up this important point. Our current theory for VecStat indeed assumes a diagonal covariance for the feature distribution, which matches the implementation where we estimate per-dimension variances and use a diagonal Gaussian for each class.
> We use this assumption due to the computational and memory constraints: Estimating and inverting a full $d\times d$ covariance per class would require $O(d^2)$ memory and preprocessing, which is prohibitive for typical representation dimensions (e.g., $d=4096$). The diagonal model keeps both storage and computation at $O(d)$ per class.
> Conceptually, extending the analysis to a full covariance is straightforward: the statistic we use can be seen as an approximation to a class-conditional Mahalanobis distance. If the true covarisance matrix $\Sigma$ is non-diagonal with eigenvalues in $[\lambda_{\min}, \lambda_{\max}]$, replacing $\Sigma^{-1}$ by a diagonal approximation corresponds to using a suboptimal quadratic form whose separation is weakened by at most a constant depending on the condition number $\kappa = \lambda_{\min}/\lambda_{\max}$.

---

> ### Author Response · Authors · 2025-11-21
>
> **Q3.How robust are these statistical methods to prompts designed to intentionally mislead the intent classifier? Given the training-free nature, are there specific vulnerabilities to adversarial attacks, and have the authors considered any countermeasures?**
>
> Thank you for raising this important question about robustness to adversarial attacks. Our initial evidence for robustness comes from the case study on ambiguous prompts (Section 3.3), where we evaluate methods on mixed-intent prompts containing both programming and mathematical content. The training-free methods (NormStat and VecStat) produce well-calibrated uncertainty distributions that appropriately reflect the prompt's mixed nature, while training-based methods exhibit overconfidence by assigning nearly all probability mass to a single class. This superior uncertainty calibration suggests that our statistical methods may be more robust to inputs that deviate from clean training data.
>
> To more rigorously evaluate adversarial robustness, we conducted an additional experiment using adversarially perturbed prompts. We created adv_math500 by prompting GPT-5-mini to inject misleading superficial features into prompts from MATH500 while preserving their mathematical content. The generation template and example pairs are provided in Tables 11 and 12 of the revised appendix, respectively. We present the results in Table 13 of the revised manuscript, and for your convenience, we also summarize the results in the following table.
>
> | Model | Method | math500 | adv_math500 |
> |-------|--------|---------|-------------|
> | **Llama-3.2-1B** | Avg-MLP | 64.33±8.33 | 1.53±1.01 |
> | | Tail-MLP | 98.93±0.42 | 94.20±2.16 |
> | | NormStat:KL | 83.40±0.00 | 77.07±0.12 |
> | | VecStat:KL | 78.80±0.12 | 12.87±0.64 |
> | | VecStat:Cos | 77.60±0.20 | 9.33±0.23 |
> | **Llama-3.2-1B-Instruct** | Avg-MLP | 72.40±2.91 | 0.73±0.12 |
> | | Tail-MLP | 94.53±0.42 | 82.33±3.06 |
> | | NormStat:KL | 96.20±0.00 | 92.40±0.00 |
> | | VecStat:KL | 87.87±0.11 | 44.20±0.69 |
> | | VecStat:Cos | 85.60±0.00 | 28.07±0.23 |
> | **Qwen3-32B** | Avg-MLP | 86.87±5.22 | 0.00±0.00 |
> | | Tail-MLP | 96.80±0.40 | 23.13±9.07 |
> | | NormStat:KL | 97.93±0.12 | 57.73±0.12 |
> | | VecStat:KL | 96.60±0.00 | 8.33±0.23 |
> | | VecStat:Cos | 96.80±0.35 | 12.20±0.53 |
>
> All methods were trained or calibrated only on clean data. We observe that their performance consistently degrades under adversarial rephrasing, underscoring the difficulty of this setting. Moreover, our results reveal several interesting patterns:
>
> **Training-based methods:** We observe divergent behaviors. Tail-MLP maintains robustness (retaining most accuracy), while Avg-MLP fails catastrophically with near-zero accuracy. This suggests that final-token representations, which attend to all previous tokens, are more resistant to scattered misleading signals than averaged embeddings.
>
> **Training-free methods:** NormStat demonstrates notable robustness, retaining 58-96% of its original accuracy under adversarial attack. For instance, on Llama-3.2-1B-Instruct, it drops only from 96.2% to 92.4%. In contrast, VecStat variants are more vulnerable (e.g., VecStat:KL drops from 87.9% to 44.2% on Llama-3.2-1B-Instruct). This aligns with our method design: NormStat's reliance on radial statistics makes it relatively invariant to directional perturbations induced by stylistic rephrasing, while VecStat's directional information makes it more sensitive to subtle changes in representation geometry caused by adversarial rephrasings.
>
> **Potential countermeasures:** Our findings suggest several strategies for improving robustness. First, combining different defense mechanisms—such as NormStat's statistical aggregation with Tail-MLP's extensive context aggregation—could provide complementary protection against adversarial attacks. Second, the superior uncertainty calibration of training-free methods could serve as an adversarial detection mechanism, flagging suspicious inputs with high uncertainty for additional scrutiny or alternative processing paths. While our training-free methods (especially NormStat) show competitive robustness, we acknowledge that achieving full adversarial robustness remains an open challenge that warrants further exploration. This analysis is now included in Appendix D.7 of the revised manuscript.

---

### Official Review · Reviewer_cfsp · 2025-11-01

**Soundness:** 2
**Presentation:** 3
**Contribution:** 3
**Rating:** 4
**Confidence:** 2

**Summary:**

This paper come up with two training-free methods based on statistical analysis of internal model representations to classify the prompt and route them to specialized models. The two methods NormStat and VecStat works well in practice while requiring negligible additional cost with theoretical guarantee.

**Strengths:**

- This paper is well-written.
- The experiment design is very fine-grained, covering many cases like ambiguous prompts, etc.

**Weaknesses:**

The motivation of this work is not very clear to me. Why is MLP not sufficient? In table 1, it mentioned that MLP needs extra calibration — I am wondering comparing to LLM pretraining, is this a bottleneck? Why do we need training-free methods in this scenario? Under what kind of situation, this method could be very useful? I think much more clear background needs to be provided in both introduction and related work. Also, MLP’s performance seems pretty nice in the experiments, which made me more confused. I assume the training samples for training a MLP is pretty easy to get, and training won’t take a long time, too.

**Questions:**

see weakness

---

> ### Author Response · Authors · 2025-11-21
>
> Dear Reviewer cfsp,
>
> Thank you for your thoughtful feedback. We have updated the manuscript accordingly, with all revisions highlighted in blue.
>
> **W1: The motivation of this work is not very clear to me. Why is MLP not sufficient? In table 1, it mentioned that MLP needs extra calibration — I am wondering comparing to LLM pretraining, is this a bottleneck? Why do we need training-free methods in this scenario? Under what kind of situation, this method could be very useful? I think much more clear background needs to be provided in both introduction and related work. Also, MLP’s performance seems pretty nice in the experiments, which made me more confused. I assume the training samples for training a MLP is pretty easy to get, and training won’t take a long time, too.**
>
> Thank you for your thoughtful comments. We would like to take this opportunity to comprehensively address your concerns about the motivation for our training-free methods compared to MLP-based approaches. Overall, the major contribution of this paper lies in providing a systematic comparison between training-free and training-based intent classification methods, and the goal is to show when one approach is better than the other.
>
> We agree that MLP methods deliver strong performance in our experiments. However, they face critical limitations in real-world deployment scenarios that motivated our work:
>
> **1. MLP methods cannot give a reasonable uncertainty quantification on prompts with mixed-intent:** As discussed in Section 3.3, VecStat/NormStat produce a reliable prediction of distribution on mixed math&code prompts, while the MLP heads are overconfident and collapse almost all mass onto a single class. In this case, we need extra effort to help calibrate the uncertainty measurement in MLP. That is why we mention that *MLP needs extra calibration* in Table 1, and this is not a bottleneck in run time, but might require extra engineering efforts to post-process the MLP's output to get a reasonable uncertainty quantification.
>
> **2. MLP methods will consume a large amount of computational resources when the number of intent classes rapidly changes:** We agree with your opinion that the training samples for training a MLP is pretty easy to get, and training won't take a long time, too. Both training-based and training-free methods require the LLM prefill stage to extract features—either for collecting statistics (for calibration) or generating training embeddings (fortraining). Thus, both approaches add negligible latency to intent classification, as their computational overhead is minimal compared to the LLM prefill phase itself. However, the fundamental difference lies in their adaptation mechanisms for evolving classification systems.
>
> Training-free methods have no trainable parameters; they require only calibration, where statistical features are collected from representative samples of each class. By contrast, training-based methods require supervised training to optimize its trainable parameters for intent discrimination. While initial training time remains trivial, practical challenges emerge in production systems that frequently incorporate new intent classes.
>
> When adding a new class, training-free methods maintain constant cost—simply computing statistics for the new class data. On the contrary, training-based methods face a trade-off: either (i) regenerate training embeddings for all classes (both existing and new) from scratch to enable retraining—a process whose cost grows linearly with the total number of classes, or (ii) cache all training embeddings to enable rapid retraining at the expense of storage. In Figure 3 of the revised manuscript, we quantify the storage implications of option (ii) for Qwen3-32B: as the number of intent classes gradually increases to 1,000, the total storage requirement reaches approximately 20GB (primarily for cached embeddings), although the MLP weights alone require only 6MB. By comparison, NormStat requires only 1.7MB for the same 1,000 classes—a difference of over an order of magnitude.
>
> These characteristics suggest that training-free methods are particularly well-suited for dynamic intent classification systems where new classes are frequently added and storage efficiency is valued, while training-based methods are only effective when the class definitions remain relatively stable and sufficient storage resources are available.

---

> ### Author Response · Authors · 2025-11-21
>
> Following the above:
>
> **Specific scenarios where our methods excel:**
>
> - High-throughput, multi-tenant systems. A provider may host a large number of routers (per product, per customer, or per domain), where the intent label space evolves over time as new tools, experts, or domains are introduced. In such settings, every change in the class set would require retraining an MLP head, whereas VecStat/NormStat only require adding or removing per-class statistics—keeping the adaptation cost essentially constant as the system scales.
>
> - Untrusted or safety-critical deployments. In systems that must handle untrusted inputs (e.g. public-facing assistants), reliable uncertainty quantification is crucial for detecting malicious or out-of-distribution prompts, and human interpretability is needed for post-hoc audits (e.g., to assess potential fairness issues). Our training-free methods directly expose calibrated per-class statistics, which can be inspected and monitored without the additional modeling and engineering complexity required to obtain well-calibrated uncertainty estimates from MLP heads. Motivated by reviewer crfd’s suggestion, in the revised manuscript we also add a new adversarial-attack experiment on MATH500 (please kindly refer to Appendix D.7 of the revised version), where we adversarially rephrase math prompts to attack the intent classifier; in this setting, all methods degrade, but NormStat retains a large fraction of its original accuracy under attack, further supporting its suitability for safety-critical routing in partially adversarial environments.
>
> We have expanded Section 2.3 and added Appendix E to clarify these motivations and use cases based on your feedback.

---

### Official Review · Reviewer_z9Cm · 2025-11-01

**Soundness:** 3
**Presentation:** 3
**Contribution:** 2
**Rating:** 2
**Confidence:** 4

**Summary:**

This paper introduces VecStat and NormStat, two "training-free" methods for classifying user intent during the prefill phase of LLM inference. These methods rely on collecting simple statistics from internal hidden layer(s) and comparing them to pre-calculated class baselines.

The authors provide a theoretical analysis to justify these statistics. Extensive empirical evaluations were conducted across multiple LLMs and two different levels of task granularity. Results were compared with lightweight supervised probes (MLPs).

**Strengths:**

Originality: this work offers a fresh perspective on feature representation analysis with respective to the intent classification problem.

Quality: this work provided rigorous theoretical grounding. With some simplification, the authors provided sound proof to supports the intuition that NormStat is good for detecting large shifts in activation magnitude (e.g. between very different domains), while VecStat retains the necessary directional information for finer-grained distinctions.

Clarity: the work is well written and easy to follow. Extensive empirical evaluations were provided.

Significance: this work offers a practical, actionable guidance for designing intent classification system.

**Weaknesses:**

Novelty is not strong enough: the application to routing is useful, but the technical innovation is relatively limited for a top-tier conference, especially considering the discriminative performance compared with peer methods. Both VecStat and NormStat work okay on easy intent classifications, but failed significantly on hard ones like Level-2 Math (I was not convinced that Level-2 Code/Natural Language is "hard".). A simple linear probe might actually have a lower memory footprint while potentially offering better discriminative performance.

Over-claim of "training free": A core selling point is that these methods are "training-free." However, they require a "calibration" phase with 2k labeled data per class. Distinguishing this "calibration" from "training" a simple probe is largely semantic.

Lack of evidence for "FAST": The computational argument against standard lightweight probes is weak. The paper lacks latency comparisons to substantiate the "FAST" claim.

Lack of strong baselines: independent routers should also be included in comparison, especially considering the labeled samples used in "calibration".

**Questions:**

1. Could you provide latency measurements comparing VecStat/NormStat against the Avg-MLP baseline? I suspect the difference is negligible relative to the attention mechanisms.

2. How does a simple linear Probe (e.g., logistic regression on z_avg) compare to the baselines? You used 2-layer MLPs, but a linear probe may be sufficient for these distinct representations and would have a lower memory footprint.

3. Could you elaborate more about the fundamental difference between calculating statistics over labeled samples vs. "training" a simple classifier?

4. Could you add one more baseline: a router-based solution? Is a simple independent router enough to solve the problem?

---

> ### Author Response · Authors · 2025-11-21
>
> Dear Reviewer z9Cm,
>
> Thank you for your thoughtful feedback. We have updated the paper accordingly, and all revisions are highlighted in blue. We believe there some misunderstandings about our contributions and the key conclusions from our work. We address the main concerns below.
>
>
> **W2:Over-claim of "training free": A core selling point is that these methods are "training-free." However, they require a "calibration" phase with 2k labeled data per class. Distinguishing this "calibration" from "training" a simple probe is largely semantic.**
>
> **W3:Lack of evidence for "FAST": The computational argument against standard lightweight probes is weak. The paper lacks latency comparisons to substantiate the "FAST" claim.**
>
> **Q1: Could you provide latency measurements comparing VecStat/NormStat against the Avg-MLP baseline? I suspect the difference is negligible relative to the attention mechanisms.**
>
> **Q3: Could you elaborate more about the fundamental difference between calculating statistics over labeled samples vs. "training" a simple classifier?**
>
> Thank you for raising these interconnected concerns about our "training-free" characterization and the "FAST" claim. We appreciate the opportunity to clarify these fundamental aspects, as they are central to understanding the practical advantages of our approach. Weakness W2 and W3 with Question Q1 and Q3 all touch on the core issue: what distinguishes our statistical "calibration" from traditional "training," and why this distinction matters for system efficiency. We address these related points together below.
>
> Both training-based and training-free methods require the LLM prefill stage to extract features—either for collecting statistics (for calibration) or generating training embeddings (for training). Thus, both approaches add negligible latency to intent classification, as their computational overhead is minimal compared to the LLM prefill phase itself. However, the fundamental difference lies in their adaptation mechanisms for evolving classification systems.
>
> Training-free methods have no trainable parameters; they require only calibration, where statistical features are collected from representative samples of each class. By contrast, training-based methods require supervised training to optimize its trainable parameters for intent discrimination. While initial training time remains trivial, practical challenges emerge in production systems that frequently incorporate new intent classes.
>
> Training-free methods maintain constant cost—simply computing statistics for the new class data. On the contrary, training-based methods face a trade-off: either (i) regenerate training embeddings for all classes (both existing and new) from scratch to enable retraining—a process whose cost grows linearly with the total number of classes, or (ii) cache all training embeddings to enable rapid retraining at the expense of storage. In Figure 3 of the revised manuscript, we quantify the storage implications of option (ii) for Qwen3-32B: as the number of intent classes gradually increases to 1,000, the total storage requirement reaches approximately 20GB (primarily for cached embeddings), although the MLP weights alone require only 6MB. By comparison, NormStat requires only 1.7MB for the same 1,000 classes—a difference of over an order of magnitude.
>
> These characteristics suggest that training-free methods are particularly well-suited for dynamic intent classification systems where new classes are frequently added and storage efficiency is valued, while training-based methods are only effective when the class definitions remain relatively stable and sufficient storage resources are available.

---

> ### Author Response · Authors · 2025-11-21
>
> **W1: Novelty is not strong enough: the application to routing is useful, but the technical innovation is relatively limited for a top-tier conference, especially considering the discriminative performance compared with peer methods. Both VecStat and NormStat work okay on easy intent classifications, but failed significantly on hard ones like Level-2 Math.**
>
>
> We believe there might be some misunderstandings on our contributions, and we would like to take this opportunity to clarify the contributions of our work. The main contribution of this paper lies in providing a **systematic comparison between training-free and training-based intent classification methods, showing when these two approaches are competitive and when one approach is better than the other**.
>
> We agree that MLP methods deliver stronger performance in our experiments, especially in level-2 classification (harder tasks). However, they face critical limitations:
>
> **1. MLP methods cannot give a reasonable uncertainty quantification on prompts with mixed-intent:** As discussed in Section 3.3, VecStat/NormStat produce a reliable prediction of distribution on mixed math&code prompts, while the MLP heads are overconfident and collapse almost all mass onto a single class. In this case, we need extra effort to help calibrate the uncertainty measurement in MLP. That is why we mention that *MLP needs extra calibration* in Table 1, and this is not a bottleneck in run time, but might require extra engineering efforts to post-process the MLP's output to get a reasonable uncertainty quantification.
>
> **2. MLP methods are not suitable for constantly changing classes:** We agree with your opinion that the training samples for training a MLP are pretty easy to get, and training won't take a long time, too. Both training-based and training-free methods require the LLM prefill stage to extract features—either for collecting statistics (for calibration) or generating training embeddings (fortraining). Thus, both approaches add negligible latency to intent classification, as their computational overhead is minimal compared to the LLM prefill phase itself. However, the fundamental difference lies in their adaptation mechanisms for evolving classification systems.
>
> On the contrary, as we mentioned in the response to the previous question, training-free methods do not suffer these disadvantages.
>
> **Specific scenarios where our methods excel:**
>
> - High-throughput, multi-router systems. A provider may host a large number of routers (per product, per customer, or per domain), where the intent label space evolves over time as new tools, experts, or domains are introduced. In such settings, every change in the class set would require retraining an MLP head, whereas VecStat/NormStat only require adding or removing per-class statistics—keeping the adaptation cost essentially constant as the system scales.
>
> - Untrusted or safety-critical deployments. In systems that must handle untrusted inputs (e.g. public-facing assistants), reliable uncertainty quantification is crucial for detecting malicious or out-of-distribution prompts, and human interpretability is needed for post-hoc audits (e.g., to assess potential fairness issues). Our training-free methods directly expose calibrated per-class statistics, which can be inspected and monitored without the additional modeling and engineering complexity required to obtain well-calibrated uncertainty estimates from MLP heads. Motivated by the reviewer’s suggestion, in the revised manuscript we also add a new adversarial-attack experiment on MATH500 (please kindly refer to Appendix D.7 of the revised version), where we adversarially rephrase math prompts to attack the intent classifier; in this setting, all methods degrade, but NormStat retains a large fraction of its original accuracy under attack, further supporting its suitability for safety-critical routing in partially adversarial environments.
>
> We have expanded Section 2.3 and added Appendix E to clarify these motivations and use cases based on your feedback.

---

> ### Author Response · Authors · 2025-11-21
>
> **Q2: How does a simple linear Probe (e.g., logistic regression on z_avg) compare to the baselines? You used 2-layer MLPs, but a linear probe may be sufficient for these distinct representations and would have a lower memory footprint.**
>
> Thank you for this suggestion. Following your recommendation, we have added a linear probe baseline (Avg-Linear) that applies logistic regression on z_avg to all experiments in the revised manuscript (Please kindly refer to Table 2-3, 7-10).
>
> The linear probe performs comparably to its 2-layer MLP counterpart (Avg-MLP) on Level-1 classification tasks. For Level-2 fine-grained classification, the linear probe achieves slightly lower accuracy than the MLP, particularly in mathematical subfield classification, suggesting that non-linearity provides some benefit for challenging within-domain discrimination tasks.
>
> We agree with your opinion that the linear probe has a lower memory footprint—$O(md)$ memory overhead versus $O(hd)$ for the MLP, where h is the MLP's hidden layer size, m is the number of classes, and d is the LLM's hidden dimension. However, in practice, this difference is negligible compared to the LLM's memory requirements during the prefill stage.
>
> While simpler, the linear probe shares the same critical limitations as other training-based methods. As demonstrated in Section 3.3, training-based methods produce overconfident predictions on ambiguous prompts, requiring post-hoc processing for reliable uncertainty estimates. In additional, the linear probe also cannot adapt to quickly evolving classification systems.
>
>
> **W4: Lack of strong baselines: independent routers should also be included in comparison, especially considering the labeled samples used in "calibration".**
>
> **Q4: Could you add one more baseline: a router-based solution? Is a simple independent router enough to solve the problem?**
>
> We thank the reviewer for this valuable suggestion. In the revision, we have added a router-based solution that uses the base LLM itself as an intent classifier through direct prompting. Table 6 presents the prompt templates used for these evaluations, and we evaluate the following two variants:
>
> - LLM Call (0-shot): Zero-shot prompting for intent classification.
> - LLM Call (3-shot): Three-shot prompting with one example per class.
>
> We found that achieving reasonable performance requires carefully designed prompts that provide clear descriptions of each intent class. Due to the complexity of designing effective prompts for fine-grained Level-2 distinctions, we evaluate this baseline only on Level-1 classification tasks. Results are presented in Tables 2 and 7 of the revised manuscript.
>
> Our experimental results reveal several key findings. First, router-based solutions require substantial model capacity—only models exceeding 4B parameters achieve performance comparable to our statistical methods. Smaller models (Llama-3.2-1B variants) fail to produce valid outputs even with in-context examples. Second, instruction fine-tuning proves critical for following classification instructions, as evidenced by the significant performance gap between Qwen3-1.7B-Base and its instruction-tuned variant. Third, achieving competitive accuracy requires carefully crafted prompts with multiple in-context examples, which substantially increases the input sequence length.
>
> These findings suggest that while an independent router can solve the intent classification problem, it introduces significant practical challenges. The requirement for larger models (≥4B parameters) and longer prompts with in-context examples directly impacts inference latency—both through increased model size and extended input sequences. In contrast, our statistical methods extract features already computed during the standard prefill phase, adding negligible overhead while achieving comparable accuracy. This makes our approach particularly suitable for latency-sensitive production systems where rapid routing decisions are critical.

---

### Official Review · Reviewer_DK5s · 2025-11-02

**Soundness:** 2
**Presentation:** 3
**Contribution:** 2
**Rating:** 2
**Confidence:** 3

**Summary:**

The paper studies LLM routing. Instead of using a trained probe, the paper proposes VecStat and NormStat based on feature statistics for intent classification, reducing the latency.
The paper provides a theoretical analysis that VecStat performs better when prompt types differ in feature directions, while NormStat is better when they differ in overall magnitude.
The paper further evaluates the proposed methods against training-based methods on LLMs, showing superior uncertainty quantification for mixed-intent prompts compared to training-based approaches, while achieving competitive accuracy.

**Strengths:**

(1) The paper proposes a more efficient method than the probe, which uses statistics to perform intent classification.

(2) The paper develops a theory to distinguish the two proposed methods, i.e., under which conditions, which method will have better performance.

(3) It is easier for the proposed method in this paper to obtain a reliable uncertainty quantification.

**Weaknesses:**

(1) In Section 2, the method section, I assume the paper should introduce the proposed method. After reading, though there are a bunch of analyses, I did not find any formulation on how the classification is performed. It's recommended to have Figure 1 to deliver the method.

(2) The paper uses baselines Avg-MLP and Tail-MLP. I'm not sure whether they are good baselines or not. Is there any existing work on this? Are the two methods good surrogates for all baselines?

(3) The accuracy in Tables 2 and 3 is most saturated, which makes the results less convincing.

**Questions:**

(1) Following Strength (2), do we know when training-based methods are better than statistical method?

---

> ### Author Response · Authors · 2025-11-21
>
> Dear Reviewer DK5s,
>
> Thank you for your thoughtful feedback. We have updated the paper accordingly, and all revisions are highlighted in blue. Below, we address each of the identified weaknesses and explain the corresponding changes made in the revised manuscript.
>
> **W1: In Section 2, the method section, I assume the paper should introduce the proposed method. After reading, though there are a bunch of analyses, I did not find any formulation on how the classification is performed. It's recommended to have Figure 1 to deliver the method.**
>
> In the revision we add a dedicated flowchart (Figure 1 in the revised version) at the start of Section 2 that summarizes the full prefill-time pipeline:
> (i) collecting module features;
> (ii) computing summary statistics;
> (iii) comparing these statistics to per-class baselines via  Gaussian KL (or cosine distance for VecStat);
> (iv) averaging distances over layers and applying a softmax over the negative distances to obtain class probabilities and a routing decision.
>
> We believe these additions will make the mechanics of VecStat and NormStat much clearer on a first read.
>
> **W2: The paper uses baselines Avg-MLP and Tail-MLP. I'm not sure whether they are good baselines or not. Is there any existing work on this? Are the two methods good surrogates for all baselines?**
>
> Our choice of Avg-MLP and Tail-MLP follows common practice in LLM-based sentence classification, where a pooled representation (average or last token) is fed into a small classifier head (e.g., Casanueva et al., 2020; Jiang et al., 2024).
>
> To further strengthen the empirical comparison, we have added additional baselines following the suggestions of Reviewer z9Cm:
>
> **Few-shot LLM-call classifier.**
>
> We have added a router-based solution that uses the base LLM itself as an intent classifier through direct prompting. Table 6 presents the prompt templates used for these evaluations, and we evaluate the following two variants:
>
> - LLM Call (0-shot): Zero-shot prompting for intent classification.
> - LLM Call (3-shot): Three-shot prompting with one example per class.
>
> We found that achieving reasonable performance requires carefully designed prompts that provide clear descriptions of each intent class. Due to the complexity of designing effective prompts for fine-grained Level-2 distinctions, we evaluate this baseline only on Level-1 classification tasks. Results are presented in Tables 2 and 7 of the revised manuscript.
>
> Our experimental results reveal several key findings. First, router-based solutions require substantial model capacity—only models exceeding 4B parameters achieve performance comparable to our statistical methods. Smaller models (Llama-3.2-1B variants) fail to produce valid outputs even with in-context examples. Second, instruction fine-tuning proves critical for following classification instructions, as evidenced by the significant performance gap between Qwen3-1.7B-Base and its instruction-tuned variant. Third, achieving competitive accuracy requires carefully crafted prompts with multiple in-context examples, which substantially increases the input sequence length.
>
> These findings suggest that while an independent router can solve the intent classification problem, it introduces significant practical challenges. The requirement for larger models (≥4B parameters) and longer prompts with in-context examples directly impacts inference latency—both through increased model size and extended input sequences. In contrast, our statistical methods extract features already computed during the standard prefill phase, adding negligible overhead while achieving comparable accuracy. This makes our approach particularly suitable for latency-sensitive production systems where rapid routing decisions are critical.
>
> **Avg-Linear baseline.**
>
> We have added a linear probe baseline (Avg-Linear) that applies logistic regression on z_avg, the average of token embeddings in the LLM's last layer, to all experiments in the revised manuscript (Please kindly refer to Table 2-3, 7-10).
>
> The linear probe performs comparably to its 2-layer MLP counterpart (Avg-MLP) on Level-1 classification tasks. For Level-2 fine-grained classification, the linear probe achieves slightly lower accuracy than the MLP, particularly in mathematical subfield classification, suggesting that non-linearity provides some benefit for challenging within-domain discrimination tasks.

---

> ### Author Response · Authors · 2025-11-21
>
> **W3: The accuracy in Tables 2 and 3 is most saturated, which makes the results less convincing.**
> We agree that many numbers in Tables 2 and 3 are close to 100%, especially for coarse-grained Level-1 tasks. However, we would like to emphasize that this saturation supports one of our main messages:
>
> For coarse-grained routing problems (math vs. code vs. text), simple statistical methods already suffice. NormStat and VecStat reach accuracy that is comparable to trained heads and LLM-call baselines, while being cheap in FLOPs and memory. This is precisely the deployment regime where one would like to adopt lightweight statistical intent classsifiers.
>
> At the same time, among out-of-distribution datasets, like math500, the accuracy is not saturated. On math500 dataset, normstat is performing better than baselines.
> In fine-grained Level-2 tasks, especially on the math-subfield benchmark, training-based heads retain a clear accuracy edge over statistical methods.
> NormStat, in particular, lags VecStat and the MLP heads on some Level-2 tasks, matching the theory that radial statistics are optimal only in isotropic-scale regimes, while directional information is needed for more subtle distinctions.
>
>
>
>
> **Q1: Following Strength (2), do we know when training-based methods are better than statistical method?**
>
> There is no single universal criterion, but our theory and experiments together give a clear rule of thumb.
>
> Conceptually, training-based heads are preferable when (i) class differences are **subtle and directional** in the representation and (iii) the intent set is **static** enough that maintaining a training and calibration pipeline is acceptable. This is exactly the regime of our hardest Level-2 tasks (especially math subfields), where MLP heads consistently outperform NormStat and VecStat by a clear margin.
>
> By contrast, statistical methods are essentially “good enough” whenever the task is **coarse-grained** and dominated by **scale or broad-domain differences** (e.g., math vs.\ code vs.\ text): on our Level-1 benchmarks, NormStat and VecStat match or nearly match MLP accuracy while offering better uncertainty and far cheaper class extendability.
>
> Empirically, we observe that training-based methods have better accuracy on fine-grained tasks, but **this comes at the expense of uncertainty quantification**: MLP heads tend to be overconfident and less expressive on mixed or ambiguous inputs, whereas our statistical methods produce more graded probability distributions that are better suited for downstream routing and fallback decisions.

---

### Note · Authors · 2025-12-24

I have read and agree with the venue's withdrawal policy on behalf of myself and my co-authors.